# Unified Enhancement of Privacy Bounds for Mixture Mechanisms via $f$-Differential Privacy

**Chendi Wang**[*]
University of Pennsylvania
Philadelphia, PA, USA, 19104 &
Shenzhen Research Institute of Big data
Shenzhen, Guangdong, China, 518000
chendi@wharton.upenn.edu

**Buxin Su**[*]
Department of Mathematics
University of Pennsylvania
Philadelphia, PA, USA, 19104
subuxin@sas.upenn.edu

**Jiayuan Ye**
Department of Computer Science
National University of Singapore
Singapore
jiayuan@comp.nus.edu.sg

**Reza Shokri**
Department of Computer Science
National University of Singapore
Singapore
reza@comp.nus.edu.sg

**Weijie J. Su**
Wharton Statistics and Data Science Department
University of Pennsylvania
Philadelphia, PA, USA, 19104
suw@wharton.upenn.edu

## Abstract

Differentially private (DP) machine learning algorithms incur many sources of randomness, such as random initialization, random batch subsampling, and shuffling. However, such randomness is difficult to take into account when proving differential privacy bounds because it induces mixture distributions for the algorithm's output that are difficult to analyze. This paper focuses on improving privacy bounds for shuffling models and one-iteration differentially private gradient descent (DP-GD) with random initializations using $f$-DP. We derive a closed-form expression of the trade-off function for shuffling models that outperforms the most up-to-date results based on $(\epsilon, \delta)$-DP. Moreover, we investigate the effects of random initialization on the privacy of one-iteration DP-GD. Our numerical computations of the trade-off function indicate that random initialization can enhance the privacy of DP-GD. Our analysis of $f$-DP guarantees for these mixture mechanisms relies on an inequality for trade-off functions introduced in this paper. This inequality implies the joint convexity of $F$-divergences. Finally, we study an $f$-DP analog of the advanced joint convexity of the hockey-stick divergence related to $(\epsilon, \delta)$-DP and apply it to analyze the privacy of mixture mechanisms.

## 1 Introduction

Differential privacy (DP, [16, 17]) is a rigorous mathematical framework for ensuring data privacy and has become a cornerstone of privacy-preserving data analysis over the past two decades. DP has found widespread applications in various data science fields, such as machine learning [12, 6, 44],

---

[*]equal contribution

37th Conference on Neural Information Processing Systems (NeurIPS 2023).

query answering [18, 15], and synthetic data generation [37, 48, 29, 28]. A randomized mechanism is considered differentially private if the outputs of two neighboring datasets that differ in at most one element are indistinguishable from each other. The closeness of these outputs can be measured in various ways, resulting in the definition of $(\epsilon, \delta)$-DP in [16] and its various relaxations.

The distinguishability between the outputs can be measured by statistical divergences. For example, $(\epsilon, \delta)$-DP is associated with the so-called hockey-stick divergence [35]. Another divergence relevant to differential privacy is the Rényi divergence [20, 38] which leads to Rényi DP [30, 10] and concentrated DP [11]. In addition to divergence-based DP, a hypothesis testing perspective on differential privacy was proposed in [43]. More recently, [14] established $f$-DP for differential privacy where the privacy is measured by the trade-off function of type I and type II errors.

In real-world applications of differential privacy, including differentially private machine learning, it is common to analyze the privacy budget of mechanisms that involve mixture distributions, where the mixture is introduced by stochastic components in the algorithm. Examples of such mechanisms include sub-sampled mechanisms [4, 52, 40, 31], shuffled mechanisms [13, 22, 23], and variants of the differentially private stochastic gradient descent (DP-SGD) algorithm [1, 9, 26, 3, 45] that involves random initialization and multiple rounds of mini-batch sampling. Recently, privacy amplification by iteration [24] has drawn much attention as it can be used to analyze the privacy bounds for DP-SGD [45, 2]) which leads to tighter privacy bounds compared to classical analysis based on the composition theorem [33, 39, 50].

While mixture mechanisms are essential in differentially private machine learning, the absence of an $f$-DP guarantee for their analysis remains a significant challenge. Moreover, existing divergence-based DP bounds for most of these mechanisms are not tight. This is primarily because the complex distribution resulting from the mixture makes it challenging to accurately quantify privacy guarantees. In order to illustrate this perspective, we consider the examples of shuffling models and DP gradient descent (DP-GD) with random initialization, as follows.

- In shuffling models, each user's data record is locally privatized using a local DP algorithm [21]. Subsequently, a curator shuffles the dataset containing all users' data. The shuffling procedure introduces additional mixtures of binomial noise [22], thereby potentially amplifying the privacy provided by the local randomizer. Shuffling is commonly employed in machine learning algorithms for batch generation [45, 47]. To deal with this mixture, Hoeffding's inequality was used in previous literature [22, 23] that leads to the loss of information. Using $f$-DP in this paper, we derive an exact analytical trade-off function for the mixture of binomial distributions which is sharp.

- In deep learning, random initialization is usually adopted in the stochastic gradient descent to enhance the performance of deep neural networks [36]. Intuitively, the inherent randomness introduced by initialization should contribute to the privacy amplification of DP-GD. However, Rényi differential privacy (DP) falls short in quantitatively measuring this randomness, even when applied to the simplest linear model. In this paper, we demonstrate how $f$-DP can effectively evaluate and quantify this inherent randomness from initialization.

**Our contributions.** This paper makes a two-fold contribution. Firstly, we propose a unified theory to analyze the privacy of mixture mechanisms within the framework of $f$-DP. Precisely, we derive an $f$-DP inequality for mixture distributions which implies the joint convexity of $F$-divergences for any convex $F$. We name this result the "joint concavity of trade-off functions", as it is a lower bound for trade-off functions. The tightness of the joint concavity is also investigated. Moreover, we propose the "advanced joint concavity of trade-off functions" which is an $f$-DP analog of the advanced joint convexity of the hockey-stick divergence and results in sharper bounds in certain cases.

Building on our inequality, we have refined the privacy analysis of both shuffling models and DP-GD with random initialization using $f$-DP. Specifically, for shuffling models, we obtain trade-off functions in a closed-form representation, leading to tighter bounds compared to existing state-of-the-art results based on $(\epsilon, \delta)$-DP. As for DP-GD, given the challenges in the trajectory analysis of multi-step iterations, we have chosen to explore a more straightforward one-iteration DP-GD. We demonstrate that using random initialization significantly enhances the privacy of the output from a single iteration.

## 2 Preliminaries on differential privacy

Let $\mathcal{D} = \{z_i\}_{i=1}^n \subset \mathcal{Z}$ be a fixed dataset of size $n$. Consider a randomized algorithm $\mathcal{A} : \mathcal{Z}^n \to \mathcal{S}$ that maps a dataset $\mathcal{D}$ to $\mathcal{A}(\mathcal{D})$ in some probability space $\mathcal{S}$. Differential privacy requires that the change of one element in a dataset has a restricted impact on the output of $\mathcal{A}$. Mathematically, we say $\mathcal{A}$ satisfies $(\epsilon, \delta)$-DP for some $\epsilon \geq 0$ and $0 \leq \delta \leq 1$ if

$$\mathbb{P}[\mathcal{A}(\mathcal{D}_0) \in S] \leq e^\epsilon \mathbb{P}[\mathcal{A}(\mathcal{D}_1) \in S] + \delta,$$

for any event $S \in \mathcal{S}$ and any neighboring datasets $\mathcal{D}_0$ and $\mathcal{D}_1$. When $\delta = 0$, we simply call $(\epsilon, 0)$-DP as $\epsilon$-DP. Based on the definition, we see that for small values of $\epsilon$ and $\delta$, it is challenging to distinguish between $\mathcal{D}_0$ and $\mathcal{D}_1$ based on the outputs of $\mathcal{A}(\mathcal{D}_0)$ and $\mathcal{A}(\mathcal{D}_1)$, as the distribution of $\mathcal{A}(\mathcal{D}_0)$ closely resembles that of $\mathcal{A}(\mathcal{D}_1)$.

The definition of $(\epsilon, \delta)$-DP corresponds to the hockey-stick divergence. Let $P$ and $Q$ be two distributions with probability density functions (pdfs) $p$ and $q$, respectively. The hockey-stick divergence between $P$ and $Q$ is defined by $H_\gamma(P\|Q) = \int (p(x) - \gamma q(x))_+ \, dx$ for $\gamma \geq 1$ with $(\cdot)_+ = \max\{0, \cdot\}$. With a little bit abuse of notations, in this paper, we define the divergence (or the trade-off function) between two random variables as the divergence (or the trade-off function) between their distributions. Then, a mechanism $\mathcal{A}$ is $(\epsilon, \delta)$-DP if and only if $H_{e^\epsilon}(\mathcal{A}(\mathcal{D}_0)\|\mathcal{A}(\mathcal{D}_1)) \leq \delta$ for any neighboring datasets $\mathcal{D}_0$ and $\mathcal{D}_1$, which also implies $H_{e^\epsilon}(\mathcal{A}(\mathcal{D}_1)\|\mathcal{A}(\mathcal{D}_0)) \leq \delta$.

The Rényi-DP (RDP) is defined based-on the Rényi divergence. The Rényi divergence of order $\widetilde{\alpha} > 1$ between $P$ and $Q$ is given by

$$R_{\widetilde{\alpha}}(P\|Q) = \frac{1}{\widetilde{\alpha} - 1} \log \int \left(\frac{p(x)}{q(x)}\right)^{\widetilde{\alpha}} q(x) dx.$$

For $\widetilde{\alpha} = 1$ or $+\infty$, $R_1$ or $R_\infty$ is the limit of $R_{\widetilde{\alpha}}$ as $\widetilde{\alpha}$ tends to 1 or $+\infty$. A mechanism $\mathcal{A}$ is said to satisfy $(\widetilde{\alpha}, \epsilon)$-RDP if $R_{\widetilde{\alpha}}(\mathcal{A}(\mathcal{D}_0)\|\mathcal{A}(\mathcal{D}_1)) \leq \epsilon$ for any neighboring $\mathcal{D}_0$ and $\mathcal{D}_1$.

The distinguishability between $\mathcal{A}(\mathcal{D}_0)$ and $\mathcal{A}(\mathcal{D}_1)$ can be quantified using hypothesis testing, which aligns with the concept of $f$-DP. Consider a hypothesis testing problem $H_0 : P$ v.s. $H_1 : Q$ and a rejection rule $\phi \in [0, 1]$. We define the type I error as $\alpha_\phi = \mathbb{E}_P[\phi]$, which is the probability that we reject the null hypothesis $H_0$ by mistake. The type II error $\beta_\phi = 1 - \mathbb{E}_Q[\phi]$ is the probability that we accept the alternative $H_1$ wrongly.

The trade-off function $T(P, Q)$ is the minimal type II error at level $\alpha$ of the type I error, that is,

$$T(P, Q)(\alpha) = \inf_\phi \{\beta_\phi : \alpha_\phi \leq \alpha\}.$$

We say a mechanism $\mathcal{A}$ satisfies $f$-DP if $T(\mathcal{A}(\mathcal{D}_0), \mathcal{A}(\mathcal{D}_1)) \geq f$ for any neighboring datasets $\mathcal{D}_0$ and $\mathcal{D}_1$. In particular, $\mathcal{A}$ is said to satisfy $\mu$-GDP if it is $G_\mu$-DP, where $G_\mu(x) = \Phi(\Phi^{-1}(1-x) - \mu)$, for $\mu \geq 0$, is the Gaussian trade-off function with $\Phi$ being the cumulative distribution function (cdf) of $\mathcal{N}(0, 1)$. $\mathcal{A}$ is considered to be more private if the corresponding trade-off function takes larger values. When $\mathcal{A}$ achieves perfect privacy and $\mathcal{A}(\mathcal{D}_0)$ and $\mathcal{A}(\mathcal{D}_1)$ become completely indistinguishable, the trade-off function is $\mathrm{Id}(x) = 1 - x$. Consequently, for any trade-off function $f$, we have $f \leq \mathrm{Id}$.

We say a trade-off function is symmetric if $T(P, Q) = T(Q, P)$. Note that a trade-off function $f$ may not necessarily be symmetric. But one can symmetrize it as shown in [14]. The symmetrization of a trade-off function will be used when we analyze the shuffled mechanisms.

## 3 Joint concavity of trade-off functions

Let $\{P_i\}_{i=1}^m$ and $\{Q_i\}_{i=1}^m$ be two sequences of probability distributions. Denote the probability density functions (pdfs) of $P_i$ and $Q_i$ as $p_i$ and $q_i$, respectively. Consider the mixture distributions $P_\mathbf{w}$ and $Q_\mathbf{w}$ with pdfs $p_\mathbf{w} = \sum_{i=1}^m w_i p_i$ and $q_\mathbf{w} = \sum_{i=1}^m w_i q_i$, where the weight $\mathbf{w} = (w_1, \cdots, w_m)$ is such that $w_i \geq 0$ and $\sum_{i=1}^m w_i = 1$. The following lemma is to bound the trade-off function $T(P_\mathbf{w}, Q_\mathbf{w})$. Upon finalizing this paper, we noted that Lemma 3.1 and Proposition 3.2 appeared independently in another paper [42, Theorem 8], where they served different applications.

**Lemma 3.1** (Joint concavity of trade-off functions). *For two mixture distributions $P_{\mathbf{w}}$ and $Q_{\mathbf{w}}$, it holds*

$$T(P_{\mathbf{w}}, Q_{\mathbf{w}})(\alpha(t,c)) \geq \sum_{i=1}^{m} w_i T(P_i, Q_i)(\alpha_i(t,c)),$$

*where $\alpha_i(t,c) = \mathbb{P}_{X \sim P_i}\left[\frac{q_i}{p_i}(X) > t\right] + c\mathbb{P}_{X \sim P_i}\left[\frac{q_i}{p_i}(X) = t\right]$ is the type I error for testing $P_i$ v.s. $Q_i$ using the likelihood ratio test and $\alpha(t,c) = \sum_{i=1}^{m} w_i \alpha_i(t,c)$.*

The main idea of the proof is to make the mixture distributions more distinguishable by releasing the indices. Precisely, for $X \sim P_{\mathbf{w}}$ and $Y \sim Q_{\mathbf{w}}$, let $X|I$ be a random variable such that $X|I = i \sim P_i$ with $I$ being the indices, i.e., $\mathbb{P}[I = i] = w_i$. Let $(X|I, I)$ be a random variable where we observe both $X|I$ and the indices $I$. Then, the right hand side of Lemma 3.1 is the trade-off function $T((X|I, I), (Y|I, I))$ between two joint distributions. This is a lower bound for the trade-off function between mixture distributions because $(X|I, I) \to X$ is a data-independent post-processing procedure that only removes the observation of indices $I$, and DP is immune to post-processing [14, 19].

Under the setting of $f$-DP, we usually require that the trade-off function is symmetric. The symmetry of the trade-off function in Lemma 3.1 is guaranteed by the following proposition.

**Proposition 3.2.** *Suppose that for each $i$, $T(P_i, Q_i)$ is a symmetric trade-off function. Then the trade-off function $T((X|I, I), (Y|I, I))$ is symmetric.*

The joint convexity of $F$-divergences plays an important role in the analysis of divergence-based DP for mixture mechanisms [4, 22]. We now show that Lemma 3.1 is an extension of the joint convexity of $F$-divergences, including the scaled exponentiation of the Rényi divergence and the hockey-stick divergence, to trade-off functions. A trade-off function is always convex and is thus differentiable almost everywhere. Thus, without loss of generality, we consider $f_i$ that is differentiable, symmetric, with $f_i(0) = 1$.

**Proposition 3.3** (An application of Lemma 3.1 to the $F$-divergences). *Let $D_F(P\|Q) = \int F(p(x)/q(x))q(x)dx$ be an $F$-divergence between any two distributions $P$ and $Q$ with some convex $F$. Then, for $f_i = T(P_i, Q_i)$, we have*

$$D_F(P_{\mathbf{w}}\|Q_{\mathbf{w}}) \leq \sum_{i=1}^{m} \int_0^1 F\left(\frac{1}{|f_i'(x)|}\right) |f_i'(x)| \, dx = \sum_{i=1}^{m} w_i D_F(P_i\|Q_i).$$

Conversion from a trade-off function to $F$-divergences is straightforward using Section B in [14]. However, conversion from an $F$-divergence to a trade-off function is highly non-trivial. In fact, $F$-divergence is an integral of a functional of the trade-off function over the whole space while Lemma 3.1 holds pointwisely, which is a local property. This explains why the divergence-based DP is not as informative as $f$-DP since some information is lost due to the integration.

## 4 Privacy analysis of the shuffled mechanisms

In this section, we explore the $f$-DP analysis of shuffled mechanisms. Drawing upon [22, 23], the shuffling procedure incorporates a mixture of binomial noise. This noise can be tightly bounded using our $f$-DP inequality for mixture distributions.

### 4.1 Theoretical privacy guarantee

In shuffling models, the record of each user is privatized by some local randomizer (such as a randomized response mechanism [41]) and all records are then shuffled by a curator. Mathematically, consider a dataset $\mathcal{D} = \{z_i\}_{i=1}^{n} \subseteq \mathcal{Z}$ of size $n$ and each data point $z_i$ is privatized by an local randomizer $\mathcal{A}_0 : \mathcal{Z} :\to \widetilde{\mathcal{Z}}$ that satisfies $\epsilon_0$-DP. Then, the mechanism $\mathcal{A} : \mathcal{Z}^n \to \widetilde{\mathcal{Z}}^n$ that maps $\mathcal{D}$ to $\widetilde{\mathcal{D}} = \{\mathcal{A}_0(z_i)\}_{i=1}^{n}$ is $\epsilon_0$-DP. A shuffler $\mathcal{A}_{\text{Shuffle}}$ takes the privatized dataset $\widetilde{\mathcal{D}}$ as input and applies a uniformly random permutation to $\widetilde{\mathcal{D}}$, which introduces the mixture of binomial noise to $\mathcal{A}$ and results in privacy amplification.

As noted in [23], the shuffling procedure introduces mixtures of binomial distributions. More specifically, the outputs generated by the shuffled mechanism for two neighboring datasets result from post-processing random variables $X \sim P$ and $Y \sim Q$ with $P = (1 - w)P_0 + wQ_0$ and $Q = (1 - w)Q_0 + wP_0$, where the weight $w = \frac{1}{e^{\epsilon_0}+1}$, and the distributions $P_0$ and $Q_0$ are defined as $(A + 1, C - A) \sim P_0$, and $(A, C - A + 1) \sim Q_0$ with $A \sim \mathrm{Binom}(C, 1/2)$ and $C \sim \mathrm{Binom}(n - 1, 2/(e^{\epsilon_0} + 1))$. It is easy to see that $P_0$ is the mixture of $\{(A_i + 1, i - A_i)\}_{i=0}^{n-1}$ with weights $w_i^0 := \mathbb{P}[C = i]$ and $Q_0$ is the mixture of $\{(A_i, i - A_i + 1)\}_{i=0}^{n-1}$ with the same weights. In this context, $\mathrm{Binom}(k, p)$ is a binomial distribution with parameters $k \in \mathbb{N}$ and $p \in [0, 1]$ and each $A_i$ is distributed as $\mathrm{Binom}(i, 1/2)$. Advancing our analysis, we adopt the joint concavity, as outlined in Lemma 3.1, to establish a lossless bound for the trade-off function $T(P_0, Q_0)$.

**Proposition 4.1.** *Let $F_i$ be the distribution function of $\mathrm{Binom}(i, 1/2)$ and let $w_i^0 = \mathbb{P}[C = i]$ for $C \sim \mathrm{Binom}(n - 1, 2/(e^{\epsilon_0} + 1))$. Then, we have $T(P_0, Q_0)$ is a piecewise linear function with*

$$T(P_0, Q_0)(\alpha(t)) = \sum_{i=0}^{n-1} w_i^0 \left\{ 1 - F_i \left[ F_i^{-1}(\alpha_i(t)) + 1 \right] \right\},$$

*for each knot $\alpha(t) = \sum_{i=0}^{n-1} w_i^0 \alpha_i(t) := \sum_{i=0}^{n-1} w_i^0 F_i \left( i - \frac{i+1}{t+1} \right)$.*

**Remark.** Proposition 4.1 holds with equality and the bound for $T(P_0, Q_0)$ is sharp.

Before stating our results for $T(P, Q)$, we define some notations related to $f$-DP. For a function $g : \mathbb{R} \to \mathbb{R}$, let $g^*(y) := \max_x \{xy - g(x)\}$ be its convex conjugate. For a trade-off function $f$, let $\mathcal{C}(f) = \min\{f, f^{-1}\}^{**}$ be its symmetrization, where $f^{-1}$ is the left inverse function of $f$, i.e., $f^{-1} \circ f(x) = x$.

**Theorem 4.2.** *The shuffled mechanism $\mathcal{A}_{\mathrm{Shuffle}} \circ \mathcal{A}$ is $\mathcal{C}(f_{\mathrm{Shuffle}})$-DP. Here $f_{\mathrm{Shuffle}}(\alpha(t))$ is a piecewise linear function where each knot $\alpha(t)$ has the form*

$$\alpha(t) = \sum_{i=0}^{n-1} w_i^0 \alpha_i(t) := \sum_{i=0}^{n-1} w_i^0 F_i \left( i - \frac{i+1}{t+1} \right) \in [0, 1], \qquad \text{for all } t \geq 0,$$

*with $F_i$ being the distribution function of $\mathrm{Binom}(i, 1/2)$ and $w_i^0 = \mathbb{P}[C = i]$ for $C \sim \mathrm{Binom}(n - 1, 2/(e^{\epsilon_0} + 1))$, and the value of $f_{\mathrm{Shuffle}}$ at a knot $\alpha(t)$ is*

$$f_{\mathrm{Shuffle}}(\alpha(t)) = 2w \cdot \mathrm{Id}(\alpha(t)) + (1 - 2w) \cdot \left[ \sum_{i=0}^{n-1} w_i^0 \left\{ 1 - F_i \left[ F_i^{-1}(\alpha_i(t)) + 1 \right] \right\} \right],$$

*with $w = \frac{1}{1+e^{\epsilon_0}}$ and $\mathrm{Id}(x) = 1 - x$ being the identity trade-off function.*

**Remark.** The bound in Theorem 4.2 is near-optimal. In fact, the proof of Theorem 4.2 is based on a post-processing procedure in [23], joint concavity (Proposition 4.1), and advanced joint concavity (Proposition 6.4). The post-processing procedure is sharp for specific mechanisms, such as the randomized response mechanism, as shown by Theorem 5.2 and Theorem 5.3 in [23]. Proposition 4.1 holds with equality and is optimal. The advanced joint concavity, which is an $f$-DP analog of the advanced joint convexity in [4], is optimal for specific distributions. Compared to existing analysis of shuffled mechanisms (e.g., [23]), the main advantage of using $f$-DP is that we avoid the use of Hoeffding's inequality and the Chernoff bound to bound the distance between $P_0$ and $Q_0$ in Proposition 4.1, which is adopted in [22, 23] and leads to loose bounds, to bound the mixture of binomial distributions. Moreover, Theorem 3.2 in [23] holds with an assumption $\epsilon_0 \leq \log \left( \frac{n}{8 \log(2/\delta)} - 1 \right)$, which is removed by using $f$-DP in our paper.

To convert $f$-DP to $(\epsilon, \delta)$-DP, we use the primal-dual perspective in [14] and obtain the following Corollary.

**Corollary 4.3.** *Let $l(t) := -\frac{\sum_{i=0}^{n-1} w_i^0 p_i \left( \lfloor i+1 - \frac{i+1}{t+1} \rfloor \right)}{\sum_{i=0}^{n-1} w_i^0 p_i \left( \lfloor i - \frac{i+1}{t+1} \rfloor \right)}$ with $p_i$ being the probability mass function of $\mathrm{Binom}(i, 1/2)$. Then, we have $\mathcal{A}_{\mathrm{Shuffle}} \circ \mathcal{A}$ is $(\epsilon, \delta_{f\text{-}DP}(\epsilon))$-DP for any $\epsilon > 0$ with*

$$\delta_{f\text{-}DP}(\epsilon) = (-e^\epsilon + 2w) \left[ \sum_{i=0}^{n-1} w_i^0 F_i \left( i - \frac{i+1}{t_\epsilon + 1} \right) \right] + (1 - 2w) \left[ \sum_{i=0}^{n-1} w_i^0 F_i \left( i + 1 - \frac{i+1}{t_\epsilon + 1} \right) \right],$$

*where $t_\epsilon = \inf\{t : -2w + (1 - 2w)l(t) \geq -e^\epsilon\}$ and $w = \frac{1}{e^{\epsilon_0}+1}$.*

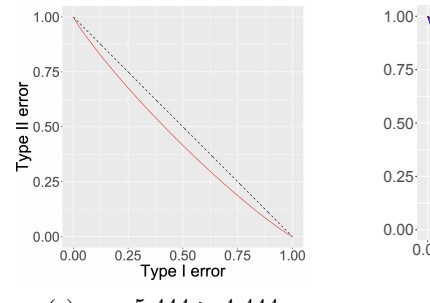 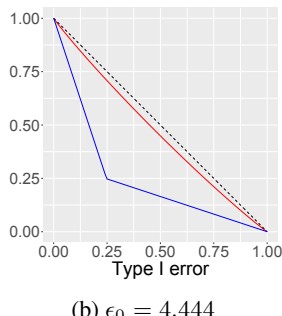 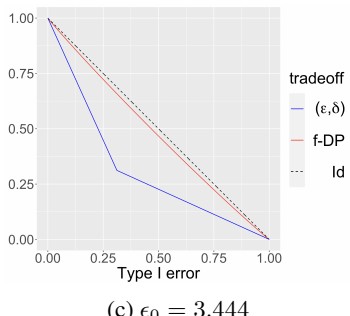

(a) $\epsilon_0 = 5.444 > 4.444$      (b) $\epsilon_0 = 4.444$      (c) $\epsilon_0 = 3.444$

Figure 1: A comparison between the trade-off function given by Theorem 4.2 and $(\epsilon, \delta)$-DP with $n = 10000$ and $\delta = n^{-1.5}$ given by [23]. [23] requires that $\epsilon_0 \leq \log\left(\frac{n}{8\log(2/\delta)} - 1\right) \approx 4.444$. Thus, there is no result for $(\epsilon, \delta)$-DP when $\epsilon_0 > 4.444$.

## 4.2 Numerical results and comparisons

To the best of our understanding, the leading privacy analysis for shuffled mechanisms is given in [23]. In this section, we compare the privacy bounds from our Theorem 4.2 and Corollary 4.3 with those found in Theorem 3.2 of [23]. Additionally, we assess the tightness of our bound against the empirical lower bounds obtained through binary search.

Specifically, Figure 1 presents a comparison of the trade-off function derived from our Theorem 4.2 to that of [23]. This comparison clearly illustrates that $f$-DP offers tighter privacy bounds, given that its trade-off function aligns closer to the identity trade-off function.

In our Table 1, we compare the values of $\delta_{f\text{-DP}}(\epsilon)$, as derived from Corollary 4.3 with $\delta(\epsilon)$ in [23]. The results indicate that $\delta_{f\text{-DP}}(\epsilon)$ is significantly smaller than $\delta(\epsilon)$.

In Table 2, we present $\epsilon_{f\text{-DP}}$ alongside the numerical upper bound of $\epsilon$ from [23] and the numerical lower bound determined by binary search. Given its closeness to the lower bound, our Theorem 4.2 can be considered near-optimal.

Table 1: Comparisons with [23]

| $\epsilon$ | 0.5 | 0.6 | 0.7 | 0.8 | 0.9 | 1.0 |
|---|---|---|---|---|---|---|
| $\delta$ in [23] | 0.9494 | 0.3764 | 0.1038 | 0.0181 | 0.0018 | $8 \times 10^{-5}$ |
| $\delta_{f\text{-DP}}$ (ours) | $3 \times 10^{-6}$ | $10^{-7}$ | $4 \times 10^{-9}$ | $9 \times 10^{-11}$ | $2 \times 10^{-12}$ | $2 \times 10^{-14}$ |

We compare $\delta_{f\text{-DP}}$ obtained in Corollary 4.3 with the corresponding $\delta$ derived from [23] using a fixed value of $\epsilon_0 = 4.444$ and $n = 10000$. Notably, $\delta_{f\text{-DP}}$ is significantly smaller than $\delta$.

Table 2: Comparisons with numerical results in [23]

| $\delta$ | $5 \times 10^{-5}$ | $3 \times 10^{-6}$ | $10^{-7}$ | $4 \times 10^{-9}$ | $9 \times 10^{-11}$ |
|---|---|---|---|---|---|
| $\epsilon_{f\text{-DP}}$ (ours) | 0.4 | 0.5 | 0.6 | 0.7 | 0.8 |
| Numerical $\epsilon$ upper bound in [23] | 1.014 | 1.085 | $\epsilon_0$ | $\epsilon_0$ | $\epsilon_0$ |
| Numerical $\epsilon$ lower bound | 0.369 | 0.470 | 0.575 | 0.664 | 0.758 |

We compare $\epsilon_{f\text{-DP}}$ obtained from Corollary 4.3 with the corresponding numerical upper bound $\epsilon$ derived from [23] using a fixed value of $\epsilon_0 = 4.444$ and $n = 10000$. For $\delta < 10^{-7}$, the bound in [23] fails as the assumption $\epsilon_0 \leq \log\left(\frac{n}{8\log(2/\delta)} - 1\right)$ is violated while our theory removes this assumption and holds for all $\epsilon_0$. Moreover, we compare our theoretical upper bound with the empirical lower bound obtained by binary search in [23] which shows that our bound is near-optimal.

In summary, our non-asymptotic privacy bound for shuffled mechanisms outperforms Theorem 3.2 in [23]. This improvement is a result of our Proposition 4.1, which optimally refines Lemma A.4 in [23]. Besides Proposition 4.1, the remainder of our proof of Theorem 4.2 closely adheres to the methodology presented in [23]. Our near-optimal result is complicated due to its tightness. Thus, it is difficult to compare our result with the asymptotic bound in [23] analytically.

## 5 Privacy analysis of one-iteration DP-GD with random initialization

A significant challenge in the privacy analysis of the last-iteration model of DP-SGD lies in accounting for multiple randomization techniques used during iterations. This includes aspects like initialization, iterative steps, and sub-sampling. Since these techniques incorporate a mixture of random noise, the joint convexity of $F$-divergence becomes crucial in the privacy analysis of DP-SGD [45, 2]. Our Lemma 3.1, which provides a unified perspective on these convexity notations, has driven us to include it in the privacy analysis of DP-GD. Nevertheless, analyzing the trajectories from multi-step iterations remains complex. Therefore, our initial exploration is to investigate the effects of random initialization on a one-step iterate. It's noteworthy that in machine learning, training a deep neural network using (stochastic) gradient descent combined with random initialization is widely adopted [36]. The significance of random initialization in noisy gradient descent is also emphasized by [46] within the framework of Kullback-Leibler privacy.

Consider a dataset $\mathcal{D} = \{(x_i, y_i)\}_{i=1}^n$ with $x_i \in \mathbb{R}$ being the features and $y_i \in \mathbb{R}$ being the labels. Let $\ell(\theta, \mathcal{D})$ be a loss function and let $g(\theta, \mathcal{D})$ be the gradient of $\ell$ with respect to $\theta$. The output of one-step iteration of DP-GD initialized at $\theta_0$ with step-size 1 is given by

$$\theta(\mathcal{D}) = \theta_0 - \left( g(\theta_0, \mathcal{D}) + \mathcal{N}(0, \sigma^2) \right). \tag{1}$$

In the setting of random initialization, $\theta_0$ is chosen as a Gaussian random variable. Without loss of generality, we consider $\theta_0 = I \sim \mathcal{N}(0, 1)$ and rewrite $\theta(\mathcal{D}) = s_I(\mathcal{D}) + \mathcal{N}(0, \sigma^2)$ with $s_I(\mathcal{D}) = I - g(I, \mathcal{D})$. $\theta(\mathcal{D})$ is a Gaussian random variable when the initialization $I$ is given, that is, $\theta(\mathcal{D})|I = i \sim \mathcal{N}(s_i(\mathcal{D}), \sigma^2)$. Thus, we can regard $\theta(\mathcal{D})$ as an infinite mixture of Gaussian distributions with continuous Gaussian weights $\{\varphi(i)\}_{i \in \mathbb{R}}$, where $\varphi$ is the pdf of $I$ and the corresponding trade-off function $T(\theta(\mathcal{D}_0), \theta(\mathcal{D}_1))$ can be bounded using the joint concavity.

For simplicity, we define $\theta(\mathcal{D})|I$ as a random variable with a given initialization $I$. For two neighboring datasets $\mathcal{D}_0$ and $\mathcal{D}_1$, it holds

$$T\left((\theta(\mathcal{D}_0)|I, I), (\theta(\mathcal{D}_1)|I, I)\right) = T\left((X|I, I), (Y|I, I)\right)$$

with $X|I \sim \mathcal{N}(0, 1)$ and $Y|I \sim \mathcal{N}(\mu_I, 1)$ for $I \sim \mathcal{N}(0, 1)$, where $\mu_I = (g(I, \mathcal{D}_1) - g(I, \mathcal{D}_0))/\sigma$.

**Theorem 5.1.** *Let $\theta(\mathcal{D}_0)$ and $\theta(\mathcal{D}_1)$ be defined in (1) for neighboring datasets $\mathcal{D}_0$ and $\mathcal{D}_1$. Then, we have*

$$T(\theta(\mathcal{D}_0), \theta(\mathcal{D}_1))(\alpha(t)) \geq \mathbb{E}_I \left[ \Phi(-t_I + \mu_I) \cdot \mathbb{1}_{[\mu_I \leq 0]} + \Phi(t_I - \mu_I) \cdot \mathbb{1}_{[\mu_I > 0]} \right]$$

*with $t_I = -\frac{t}{\mu_I} + \frac{\mu_I}{2}$ and $\alpha(t) = \mathbb{E}_I \left[ \Phi(t_I) \cdot \mathbb{1}_{[\mu_I \leq 0]} + \Phi(-t_I) \cdot \mathbb{1}_{[\mu_I > 0]} \right]$. Here $\Phi$ is the cumulative distribution function of $\mathcal{N}(0, 1)$ and the expectation is taken with respect to $I$.*

**Remark.** Note that Theorem 5.1 is instance-based privacy guarantee as it relies on the datasets. To extend it to the worst case, we let $\mu_I^{\max} = \max_{\mathcal{D}_0, \mathcal{D}_1} \left\{ |g(I, \mathcal{D}_1^{\max}) - g(I, \mathcal{D}_0^{\max})| / \sigma \right\}$ be the sensitivity of the gradient with a given initialization $I$. As a result, $\theta(\mathcal{D})$ output by one-step DP-GD is $f$-DP with $f(\alpha(t)) = \mathbb{E}_I \left[ \Phi(t_I^{\max} - \mu_I^{\max}) \right]$, where $t_I^{\max} = -\frac{t}{\mu_I^{\max}} + \frac{\mu_I^{\max}}{2}$ and $\alpha(t) = \mathbb{E}_I \left[ \Phi(-t_I^{\max}) \right]$. The worst case trade-off function is bounded for strongly convex loss functions with a bounded data domain.

To numerically evaluate the trade-off function in Theorem 5.1, we consider an example $\mathcal{D}_0 = \{(x_i, y_i)\}_{i=1}^n$ with $y_i = a x_i$ and $x_i^2 = 1$ for some constant $a$ and we defined $\mathcal{D}_1$ by removing an arbitrary element in $\mathcal{D}_0$. Moreover, we assume that $\sigma = 1$. Note that for this example without gradient clipping, the gradient is linear in $I$ and $\theta(\mathcal{D}_0)$ is the sum of two Gaussian random variables which is Gaussian. Thus, the trade-off function has a closed-form representation. In general, the output is non-Gaussian and we should adopt Theorem 5.1. For example, if we consider gradient clipping [1, 9] and replace $g(\theta, \mathcal{D})$ by the clipped gradient

$$g_c(\theta, \mathcal{D}) = \sum_{i=1}^n \frac{g^{(i)}(\theta)}{\max\{1, \|g^{(i)}(\theta)\|_2 / c\}}, \qquad \text{with } g^{(i)}(\theta) = (y_i - \theta x_i)(-x_i),$$

where the gradient of each data point $g^{(i)}$ is cut off by some constant $c > 0$, then $\mu_I^{\max}$ is given by

$$\mu_I^{\max} = \begin{cases} a - I, & |a - I| \leq c, \\ c, & a - I \geq c, \\ -c, & a - I \leq -c, \end{cases}$$

which is not Gaussian. In this example $g_c(\theta, \mathcal{D}) + \mathcal{N}(0, 1)$ is considered as $c$-GDP if we disregard the effects of random initialization since the sensitivity of $g_c$ is $c$.

We illustrate the trade-off function of Theorem 5.1 computed numerically in Figure 2, where we also compare it with $c$-GDP for $a = 1$ and varying values of $c$. Overall, the figure suggests that random initialization can amplify the privacy of DP-GD, as our bounds outperform those of $c$-GDP, which does not take into account the randomness of initialization. Furthermore, we observe that as $c$ increases, the amplification effect caused by random initialization becomes more significant, since the difference between $T((X|I, I), (Y|I, I))$ and $c$-GDP also increases. This is reasonable, since the randomness resulting from initialization comes from $I$ such that $|a - I| \leq c$, whereas for $|a - I| > c$, $\mu_I$ remains constant and no randomness is introduced. Thus, the random initialization introduces greater levels of randomness as $c$ increases.

It is worth noting that in this example, without gradient clipping, we have $\mu_I^{\max} = a - I$ and the dominate pair are two Gaussian distributions $\mathcal{N}(0, 1)$ and $\mathcal{N}(0, 2)$. The Rényi DP fails to measure the privacy of initialization. In fact, it holds $R_{\widetilde{\alpha}}(\mathcal{N}(0, 1) \| \mathcal{N}(0, 2)) = \infty$ for $\widetilde{\alpha}$ large enough.

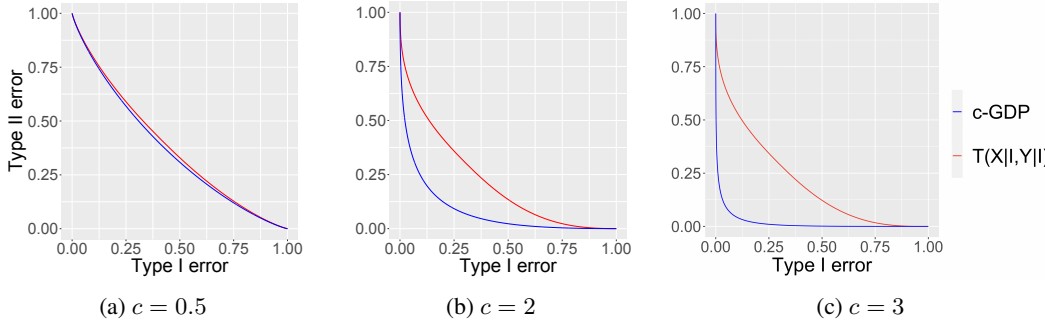

| (a) $c = 0.5$ | (b) $c = 2$ | (c) $c = 3$ |

Figure 2: Trade-off functions for linear models with $a = 1$.

## 6 Optimality of joint concavity and advanced joint concavity

In this section, we first explore the sufficient and necessary conditions under which Lemma 3.1 holds with equality. While Lemma 3.1 is generally not sharp, we introduce an $f$-DP analog of the advanced joint convexity of the hockey-stick divergence from [4], yielding tighter bounds in certain applications.

Recall the distributions $P = (1 - w)P_0 + wQ_0$ and $Q = (1 - w)Q_0 + wP_0$ that appear in the shuffled mechanisms. Bounding the trade-off function $T(P, Q)$ directly using the joint concavity leads to a loose bound (cf., Figure 3b). For the scenarios where Lemma 3.1 is not tight, we introduce the $f$-DP analog of the advanced joint convexity of $(\epsilon, \delta)$-DP [4] that may lead to tighter bounds and we term it the "advanced joint concavity of trade-off functions".

The following proposition presents a necessary and sufficient condition for Lemma 3.1 to hold with equality.

**Proposition 6.1.** *For $m = 2$, Lemma 3.1 holds with equality if and only if $\frac{w_1 p_1 + w_2 p_2}{w_1 q_1 + w_2 q_2}(X) \overset{\mathbb{P}}{=} w_1 \frac{p_1}{q_1}(X) + w_2 \frac{p_2}{q_2}(X)$ with $X \sim P_{\mathbf{w}}$, where for $p_i(X)/q_i(X) = 0/0$ and $p_j(X)/q_j(X) \neq 0/0$ with $i \neq j$, we set $p_i(X)/q_i(X) = p_j(X)/q_j(X)$.*

It is not difficult to see that $P_0$ and $Q_0$ in shuffling models satisfy this necessary and sufficient condition when $n = 2$.

As we discussed, Lemma 3.1 may not be sharp in general. The following lemma is about the advanced joint convexity of the hockey-stick divergence, which is a slight generalization of Theorem 2 in [4].

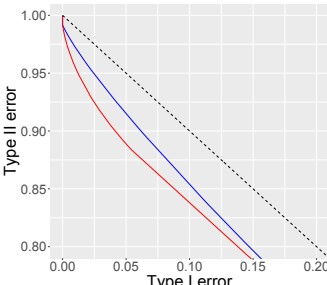
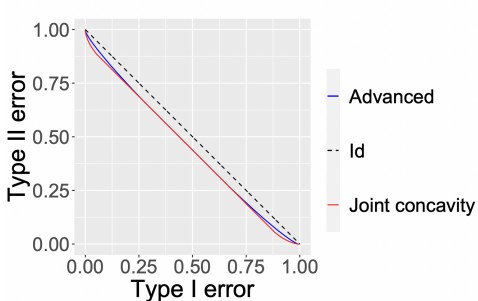

(a) Top-left corner of Figure 3b for small type I error $\alpha$. As the parameters $(\epsilon, \delta)$ computed by the trade-off function only depend on small $\alpha$, advanced joint concavity leads to a tighter bound.

(b) Trade-off functions obtained by (advanced) joint concavity. For example, for $w = 1/3$ and $\epsilon = 0.5$, $\delta$ derived from Lemma 6.3 is $1.5 \times 10^{-6}$ while that from Lemma 3.1 is 0.0020.

Figure 3: Comparison between joint concavity (Lemma 3.1) and advanced joint concavity (Lemma 6.3).

**Lemma 6.2.** *For any non-negative $\epsilon', \epsilon_0, \epsilon_1, \gamma$, and $\eta$ satisfying $\exp(\epsilon') = (1-w)\exp(\epsilon_0) + w\exp(\epsilon_1)$ and $\exp(\epsilon_0)(1-w)\gamma + \exp(\epsilon_1)w\eta = \exp(\epsilon')w$, we have*

$$
\begin{aligned}
H_{e^{\epsilon'}} & ((1-w)P_1 + wP_2 \| (1-w)Q_1 + wQ_2) \\
& \leq (1-w)H_{e^{\epsilon_0}}(P_1 \| (1-\gamma)Q_1 + \gamma Q_2) + wH_{e^{\epsilon_1}}(P_2 \| (1-\eta)Q_1 + \eta Q_2).
\end{aligned}
\tag{2}
$$

Lemma 6.2 is reduced to the advanced joint convexity of the hockey-stick divergence in [4] when $P_1 = Q_1$, by minimizing the right-hand-side of (2) with respect to $\gamma, \eta, \epsilon_0$, and $\epsilon_1$.

Recall the convex conjugate $g^*$ of a function $g$ defined by $g^*(y) = \sup_x\{xy - g(x)\}$ and $\mathcal{C}(f) = \min\{f, f^{-1}\}^{**}$ which is the symmetrization of $f$. We have the following advanced joint concavity of trade-off functions.

**Lemma 6.3** (Advanced joint concavity). *Suppose that $T(P_i, Q_i)$ is symmetric for each $i$. Then, for $0 \leq w \leq 1$, we have*

$$
\begin{aligned}
T((1-w)P_1 + wP_2, (1-w)Q_1 + wQ_2) \\
\geq \mathcal{C}\left(\left((1-w)(1-\gamma)F_{1,1}^* + w(1-\eta)F_{2,1}^* + (1-w)\gamma F_{1,2}^* + w\eta F_{2,2}^*\right)^*\right)
\end{aligned}
$$

*for arbitrary $0 \leq \gamma < w < \eta \leq 1$, where $F_{i,j}(x)$ is given by $F_{1,i}(x) := f_{1,i}\left(\frac{x(1-w)(\eta-\gamma)}{(\eta-w)}\right)$, and $F_{2,i}(x) := f_{2,i}\left(\frac{xw(\eta-\gamma)}{(w-\gamma)}\right)$, and the trade-off functions are defined as $f_{i,j} = T(P_i, Q_j)$ for $1 \leq i, j \leq 2$. Moreover, for $\gamma = \eta = w$, it holds*

$$
\begin{aligned}
T((1-w)P_1 + wP_2, (1-w)Q_1 + wQ_2) \\
\geq \mathcal{C}\left((1-w)T(P_1, (1-w)Q_1 + wQ_2) + wT(P_2, (1-w)Q_1 + wQ_2)\right).
\end{aligned}
$$

Determining the trade-off functions using advanced joint concavity can be challenging in many practical situations. In fact, to apply the advanced joint concavity, one need to specify the choice of $\gamma, \eta$ by maximizing the right-hand-side of Lemma 6.3. Therefore, in real-world applications, we often rely on both joint concavity and advanced joint concavity.

For $P = (1-w)P_0 + wQ_0$ and $Q = (1-w)Q_0 + wP_0$ in shuffling models, we have the following bound derived from Lemma 6.3.

**Proposition 6.4.** *For $P = (1-w)P_0 + wQ_0$ and $Q = (1-w)Q_0 + wP_0$ with some weight $0 \leq w \leq 1/2$, we have $T(P, Q) \geq \mathcal{C}(2w\mathrm{Id} + (1-2w)T(P_0, Q_0))$.*

The equality in Proposition 6.4 does not hold exactly. However, this lower bound is almost the tightest closed-form expression. One may refer to Section E.1.1 in the appendix for the proof details.

# 7  Discussion

This paper provides refined privacy bounds for mixture mechanisms, including shuffling models and DP-GD with random initialization. For shuffling models, we present a bound that is tighter than existing results based on $(\epsilon, \delta)$-DP. In the study of DP-GD, we demonstrate how random initialization can amplify privacy concerns. These bounds are derived using a unified $f$-DP approach based on the joint concavity and advanced joint concavity of trade-off functions. We also investigate the sharpness and other properties of these concavity notions.

In our future work, we plan to extend our analysis from one-step DP-GD to multi-step DP-SGD. For DP-SGD with multiple iterations, it is crucial to consider subsampling and privacy amplification by iteration in the privacy accountant, in addition to the randomness introduced by shuffling and random initialization. While there is an $f$-DP bound for subsampling provided in an independent work [42], as far as we know, there is limited research on $f$-DP results regarding privacy amplification by iteration.

Beyond DP-SGD, we intend to extend our theory to the privacy analysis of other key applications that involves various randomization techniques. These include the shuffled Gaussian mechanism for federated learning, as discussed in [25], and the composition of mixture mechanisms. For extending our theory to federated learning, we might adopt the $f$-DP framework outlined in [49]. Addressing the composition of mixture mechanisms demands examination of the tensor product between trade-off functions. This is a complex task, even when dealing with the simplest mixture mechanisms like sub-sampling, as highlighted in [51].

## Acknowledgments

Weijie J. Su was supported in part by a Meta Research Award and NSF through CCF1934876.

Reza Shokri was supported by a Google PDPO Faculty Research Award, Intel within the www.private-ai.org center, a Meta Faculty Research Award, the NUS Early Career Research Award (NUS ECRA award number NUS ECRA FY19 P16), and the National Research Foundation, Singapore under its Strategic Capability Research Centres Funding Initiative.

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
