# A  Essential foundations of differential privacy

The Neyman-Pearson lemma (cf., [27]) is crucial in the proof as it establishes the likelihood ratio test as the most powerful test.

**Lemma A.1** (Neyman-Pearson). *Let $P$ and $Q$ be probability distributions on $\Omega$ with densities $p$ and $q$, respectively. For the hypothesis testing problem $H_0 : P$ vs $H_1 : Q$, a test $\phi : \Omega \to [0,1]$ is the most powerful test at level $\alpha$ if and only if there are two constants $t \in [0, +\infty]$ and $c \in [0,1]$ such that $\phi$ has the form*

$$\phi(\omega) = \begin{cases} 1, & \text{if } p(\omega) < tq(\omega), \\ c, & \text{if } p(\omega) = tq(\omega), \\ 0, & \text{if } p(\omega) > tq(\omega). \end{cases}$$

As an application of the Neyman-Pearson lemma, the type I error $\alpha(t)$ has the form

$$\alpha(t) = \mathbb{E}_P[\phi] = \mathbb{P}_{X \sim P}\left[\frac{p(X)}{q(X)} < t\right] + c\mathbb{P}_{X \sim P}\left[\frac{p(X)}{q(X)} = t\right],$$

and the type II error is

$$\beta(t) = 1 - \mathbb{E}_Q[\phi] = \mathbb{P}_{X \sim Q}\left[\frac{p(X)}{q(X)} > t\right] + (1-c)\mathbb{P}_{X \sim Q}\left[\frac{p(X)}{q(X)} = t\right].$$

One of the most important properties of differential privacy is that DP is immune to data-independent post-processing. Precisely, we introduce the following information processing inequality given by [14].

**Lemma A.2** (Theorem 2.10 in [14]). *Let $P$ and $Q$ be two distributions on a probability space $\mathcal{Z}$ and let $\widetilde{P}$ and $\widetilde{Q}$ be two distributions on another probability space $\widetilde{\mathcal{Z}}$. The following two statements are equivalent:*

*(a) $T(P,Q) \leq T(\widetilde{P}, \widetilde{Q})$.*

*(b) There exists a post-processing algorithm $\mathrm{Proc} : \mathcal{Z} \to \widetilde{\mathcal{Z}}$ such that $\mathrm{Proc}(P) = \widetilde{P}$ and $\mathrm{Proc}(Q) = \widetilde{Q}$.*

The primal-dual perspective, initially introduced by [14], will be employed to explore the relationship between $(\epsilon, \delta)$-DP and $f$-DP. Recall that for a function $g$, its convex conjugate $g^*$ is defined by $g^*(y) = \sup_x\{xy - f(x)\}$.

**Lemma A.3** (Proposition 2.12 in [14]). *Let $f$ be a symmetric trade-off function. A mechanism is $f$-DP if and only if it is $(\epsilon, \delta(\epsilon))$-DP for all $\epsilon > 0$ with $\delta(\epsilon) = 1 + f^*(-e^\epsilon)$.*

To make use of Lemma 6.3, we recall the symmetrization of a trade-off function as defined in Definition F.1 of [14]. Let $f$ be a trade-off function, the symmetrization of $f$ is given by

$$\mathrm{Symm}(f) = \begin{cases} \min\{f, f^{-1}\}^{**}, & \text{if } \bar{x} \leq f(\bar{x}), \\ \max\{f, f^{-1}\}, & \text{if } \bar{x} > f(\bar{x}), \end{cases}$$

with $\bar{x} = \inf\{x \in [0,1] : -1 \in \partial f(x)\}$.

According to Section F in [14], we have

$$\min\{f, f^{-1}\}^{**} = \begin{cases} f(x), & 0 \leq x \leq \bar{x}, \\ \bar{x} - f(\bar{x}) - x, & \bar{x} \leq x \leq f(\bar{x}), \\ f^{-1}(x), & f(\bar{x}) \leq x \leq 1. \end{cases} \tag{3}$$

Another useful tool is the advanced joint convexity first introduced in [4].

**Lemma A.4** (Theorem 2 in [4]). *Let $P$ and $Q$ be two distributions such that $P = (1-w)P_0 + wP_1$ and $Q = (1-w)P_0 + wQ_1$, for some $0 \leq w \leq 1$. Given $\gamma \geq 1$, let $\gamma' = 1 + w(\gamma - 1)$ and $\eta = \gamma'/\gamma$. Then, it holds*

$$H_{\gamma'}(P\|Q) = wH_\gamma(P_1\|(1-\eta)P_0 + \eta Q_1).$$

# B Technical details of Section 3

In this section, we discuss the omitted details of Section 3. Prior to delving into the proofs, we provide a reminder of the notations. Let $\{P_i\}_{i=1}^m$ and $\{Q_i\}_{i=1}^m$ be two sequences of probability distributions. For a weight vector $\mathbf{w} = (w_1, \cdots w_m)$, let $P_{\mathbf{w}} = \sum_{i=1}^m w_i P_i$ and let $Q_{\mathbf{w}} = \sum_{i=1}^m w_i Q_i$. Let $I$ be a random variable such that $\mathbb{P}[I = i] = w_i$.

## B.1 Proof of Lemma 3.1 and discussions

*Proof of Lemma 3.1.* Consider $X \sim P_{\mathbf{w}}$ and $Y \sim Q_{\mathbf{w}}$. Here, $(X|I, I)$ denotes the observation of $X|I = i \sim P_i$ along with an index $i$, indicating that $X$ is drawn from the $I$-th distribution $P_I$. Therefore, $(X|I, I) \to X$ represents a post-processing step where we remove the information about $I$. Since we solely manipulate the indices, this post-processing is independent of the data, leading to the inequality $T(X, Y) \geq T((X|I, I), (Y|I, I))$.

The next step is to specify $T((X|I, I), (Y|I, I))$. Let $p_I$ and $q_I$ be the pdfs of $P_I$ and $Q_I$, respectively. According to Lemma A.1, the most powerful test is the likelihood ratio test. Then, the type I error is

$$\alpha(t, c) = \mathbb{P}_{X \sim P_I, I}\left[\frac{p_I(X)}{q_I(X)} < t\right] + c\mathbb{P}_{X \sim P_I, I}\left[\frac{p_I(X)}{q_I(X)} = t\right]$$

$$= \mathbb{E}_I\left[\mathbb{P}_{X \sim P_I}\left[\frac{p_I(X)}{q_I(X)} < t \,\middle|\, I\right] + c\mathbb{P}_{X \sim P_I}\left[\frac{p_I(X)}{q_I(X)} = t \,\middle|\, I\right]\right]$$

$$= \sum_{i=1}^m w_i \alpha_i(t, c),$$

where

$$\alpha_i(t, c) = \mathbb{P}_{X \sim P_I}\left[\frac{p_I(X)}{q_I(X)} < t \,\middle|\, I = i\right] + c\mathbb{P}_{X \sim P_I}\left[\frac{p_I(X)}{q_I(X)} = t \,\middle|\, I = i\right]$$

$$= \mathbb{P}_{X \sim P_i}\left[\frac{p_i(X)}{q_i(X)} < t\right] + c\mathbb{P}_{X \sim P_i}\left[\frac{p_i(X)}{q_i(X)} = t\right].$$

Similarly, the type II error is

$$\beta(t, c) = \mathbb{E}_I\left[\mathbb{P}_{X \sim Q_I}\left[\frac{p_I(X)}{q_I(X)} > t \,\middle|\, I\right] + (1 - c)\mathbb{P}_{X \sim Q_I}\left[\frac{p_I(X)}{q_I(X)} = t \,\middle|\, I\right]\right]$$

$$= \sum_{i=1}^m w_i \left(\mathbb{P}_{X \sim Q_i}\left[\frac{p_i(X)}{q_i(X)} > t\right] + (1 - c)\mathbb{P}_{X \sim Q_i}\left[\frac{p_i(X)}{q_i(X)} = t\right]\right)$$

$$=: \sum_{i=1}^m w_i \beta_i(t, c).$$

We complete the proof by noting that $\beta_i = T(P_i, Q_i)(\alpha_i)$. $\qquad\square$

**Remark.** Lemma 3.1 can be extended to continuous weights $I$. In fact, for $I$ being a random variable with pdf $\varphi$, one still has $T(X, Y) \geq T((X|I, I), (Y|I, I))$ using the same post-processing. Then, we have

$$\alpha(t, c) = \mathbb{E}_I\left[\mathbb{P}_{X \sim P_I}\left[\frac{p_I(X)}{q_I(X)} < t \,\middle|\, I\right] + c\mathbb{P}_{X \sim P_I}\left[\frac{p_I(X)}{q_I(X)} = t \,\middle|\, I\right]\right] \tag{4}$$

and

$$\beta(t, c) = \mathbb{E}_I\left[\mathbb{P}_{X \sim Q_I}\left[\frac{p_I(X)}{q_I(X)} > t \,\middle|\, I\right] + (1 - c)\mathbb{P}_{X \sim Q_I}\left[\frac{p_I(X)}{q_I(X)} = t \,\middle|\, I\right]\right], \tag{5}$$

which are non-elementary integrals. This continuous analog of Lemma 3.1 will be used to prove Theorem 5.1.

Lemma 3.1 can be extended to $P_{\mathbf{w}}$ and $Q_{\mathbf{w}'}$ with different weights $\mathbf{w}$ and $\mathbf{w}'$.

**Proposition B.1.** *Let* $\mathbf{w} = (w_1, w_2)$ *and* $\mathbf{w}' = (w_1', w_2')$. *It holds*

$$T(P_{\mathbf{w}}, Q_{\mathbf{w}'}) \geq \min\{w_1, w_1'\}T(P_1, Q_1) + \min\{w_2, w_2'\}T(P_2, Q_2)$$
$$+ (w_2' - \min\{w_2, w_2'\})T(P_1, Q_2) + (w_1' - \min\{w_1, w_1'\})T(P_2, Q_1).$$

*In addition, similar results hold for any* $m \geq 2$.

## B.2 Characterization of $T((X|I, I), (Y|I, I))$ and proof of Proposition 3.2

The type I error $\alpha_i(t)$ can be represented by the trade-off function $T(P_i, Q_i)$ and the rejection region decided by $t$ and $c$. For simplicity, in the following of this section, we only discuss the continuous case with $\mathbb{P}[q/p = t] = 0$, where the trade-off function is differentiable, and rewrite $\alpha_i(t, c) = \alpha_i(t)$. Precisely, we have the following proposition.

**Proposition B.2.** *Let* $T(P_i, Q_i) = f_i$ *with some differentiable trade-off function* $f_i$. *Suppose that*

$$\int_{\mathbb{R}} \delta\left(t - \frac{q_i(x)}{p_i(x)}\right) q_i(x)dx = \int_{\frac{q_i(x)}{p_i(x)} = t} \frac{q_i(x)}{D\left(\frac{q_i}{p_i}\right)(x)} dx < \infty, \tag{6}$$

*for any* $t > 0$ *and* $1 \leq i \leq m$, *where* $\delta$ *is the Dirac delta function and* $D(\frac{q_i}{p_i})(x)$ *is the weak derivative of* $q_i/p_i$. *Then, we have*

$$\alpha(t) = \sum_{i=1}^{m} w_i \left(f_i'\right)^{-1}(-t) \quad \text{and} \quad T(P_I, Q_I)(\alpha(t)) = \sum_{i=1}^{m} w_i f_i\left((f_i')^{-1}(-t)\right). \tag{7}$$

The key observation from Proposition B.2 is that

$$\left.\frac{df_i}{d\alpha_i}\right|_{\alpha_i = \alpha_i(t)} = -t \quad \text{and} \quad \left.\frac{dT(X|I, Y|I)(\alpha)}{d\alpha}\right|_{\alpha = \alpha(t)} = -t,$$

which means $f_i$ and $T(P_I, Q_I)$ have the same derivative at points $\alpha_i(t)$ and $\alpha(t)$ induced by the same threshold $t$. This observation is important in our analysis of the applications to shuffling models and to derive the joint convexity of $F$-divergences. Equation (6) is not a strong assumption. For example, when $q_i/p_i$ is strictly monotone, we have $\delta(t - q_i/p_i(x)) = \delta(x - x_t)$ with $q_i/p_i(x_t) = t$ and

$$\int_{\mathbb{R}} \delta\left(t - \frac{q_i(x)}{p_i(x)}\right) q_i(x)dx = q_i(x_t) \leq 1.$$

**Corollary B.3.** *Let* $T(P_i, Q_i) = f_i$ *with some differentiable trade-off function* $f_i$. *Suppose that* $p_i/q_i$ *is monotone. Then, we have* $\alpha(t) = \sum_{i=1}^{m} w_i (f_i')^{-1}(-t)$ *and* $T(P_I, Q_I)(\alpha(t)) = \sum_{i=1}^{m} w_i f_i\left((f_i')^{-1}(-t)\right)$.

*Proof of Proposition B.2.* Rewrite $\alpha_i(t)$ and $\beta_i(t) = f_i(\alpha_i(t))$ as

$$\alpha_i(t) = \int_{\mathbb{R}} \mathbb{1}\left[q_i(x)/p_i(x) < t\right] p_i(x)dx = \int_{\mathbb{R}} \mathbb{1}\left[q_i(x)/p_i(x) < t\right] \frac{p_i(x)}{q_i(x)} q_i(x)dx$$

and

$$\beta_i(t) = \int_{\mathbb{R}} \mathbb{1}\left[p_i(x)/q_i(x) > t\right] q_i(x)dt.$$

Then we have

$$\frac{d\alpha_i}{dt} = \int_{\mathbb{R}} -\delta\left(t - \frac{q_i}{p_i}(x)\right) \frac{p_i}{q_i}(x) q_i(x)dx,$$

where $\delta$ is the Dirac delta function. Note that

$$\delta\left(t - \frac{q_i}{p_i}(x)\right) \frac{p_i}{q_i}(x) = \frac{\delta\left(t - \frac{q_i}{p_i}(x)\right)}{t}.$$

We get

$$\frac{d\alpha_i}{dt} = -\frac{1}{t} \int_{\mathbb{R}} \delta\left(t - \frac{q_i}{p_i}(x)\right) q_i(x) dx.$$

Since

$$\frac{d\beta_i}{dt} = \int_{\mathbb{R}} \delta\left(t - \frac{q_i}{p_i}(x)\right) q_i(x) dx,$$

we obtain

$$f_i'(\alpha_i) = \frac{d\beta_i}{d\alpha_i} = \frac{d\beta_i}{dt}\frac{dt}{d\alpha_i} = -t$$

and $\alpha_i = (f_i')^{-1}(-t)$. $\qquad\square$

Based on the proof of Proposition B.2, we now prove Proposition 3.2.

*Proof of Proposition 3.2.* It is sufficient to show that, for any $(\alpha(t), \beta(t))$ such that $\beta(t) = f(\alpha(t))$, there is a $\widetilde{t}$ such that $\beta(\widetilde{t}) = \alpha(t) = f(\alpha(\widetilde{t}))$ with

$$\alpha(\widetilde{t}) = \sum_{i=1}^{m} w_i \alpha_i(\widetilde{t}) = \sum_{i=1}^{m} w_i \beta_i(t).$$

As $f_i$ is symmetric, for any $1 \le i \le m$, there is $\widetilde{t}_i$ such that $\alpha_i(\widetilde{t}_i) = \beta_i(t)$ and $\beta_i(\widetilde{t}_i) = \alpha_i(t)$. So, it is enough to show that $\widetilde{t}_i = \widetilde{t}$ for all $i$. Recall that in the proof of Proposition B.2, we have

$$\left.\frac{d\beta_i}{d\alpha_i}\right|_{\alpha_i = \alpha_i(t)} = -t.$$

Now we consider $d\beta_i(\widetilde{t}_i)/d\alpha_i(\widetilde{t}_i) = -\widetilde{t}_i$. On the other hand, since

$$\beta_i(\widetilde{t}_i) = f_i(\alpha_i(\widetilde{t}_i)) = f_i^{-1}(\alpha_i(\widetilde{t}_i)),$$

using the inverse funciton theorem, we have

$$d\beta_i(\widetilde{t}_i)/d\alpha_i(\widetilde{t}_i) = \frac{1}{f_i'(f_i^{-1}(\alpha_i(\widetilde{t}_i)))} = \frac{1}{f_i'(\alpha_i(t))} = -\frac{1}{t},$$

where the second equality is because $\alpha_i(\widetilde{t}_i) = \beta_i(t) = f_i(\alpha_i(t))$ and the third equality is from $\alpha_i(t) = (f_i')^{-1}(-t)$ that appears in Proposition B.2. Overall, we obtain $\widetilde{t}_i = \frac{1}{t}$ for all $i$. $\qquad\square$

### B.3   Conversion from Lemma 3.1 to $F$-divergences

In this section, we investigate the relationship between Lemma 3.1 and $F$-divergences. For two distributions $P$ and $Q$, the $F$-divergence between $P$ and $Q$ is given by

$$D_F(P\|Q) = \int_{pq>0} F(p/q) dQ + F(0)\mathbb{P}_Q[p=0] + \tau_F \cdot \mathbb{P}_P[q=0],$$

where $F(0) = \lim_{s\to 0} F(s)/s$ and $\tau_F = \lim_{t\to\infty} F(t)/t$.

To convert Lemma 3.1 to $F$-divergences, we recall the relationship between trade-off functions and $F$-divergences in [14]. Precisely, for any $F$-divergence $D_F(P\|Q)$ between two distributions $P$ and $Q$, there is a functional $l_F(T(P,Q))$ such that $D_F(P\|Q) = l_F(T(P,Q))$. This functional $l_F$ can be specified using the following lemma.

**Lemma B.4** (Proposition B.4 in [14]). *Let $z_f := \inf\{x \in [0,1], f(x) = 0\}$ be the first zero of a trade-off function $f$. The functional $l_F$ that computes the $F$-divergence $D_F$ has the following expression*

$$l_F(f) = \int_0^{z_f} F\left(\frac{1}{|f'(x)|}\right) \cdot |f'(x)| dx + F(0)(1 - f(0)) + \tau_F(1 - z_f).$$

*In particular, if $f$ is symmetric with $f(0) = 1$, then we have*

$$l_F(f) = \int_0^{z_f} F\left(\frac{1}{|f'(x)|}\right) \cdot |f'(x)| dx.$$

Now we prove Proposition 3.3.

*Proof of Proposition 3.3.* Let $f(\alpha) := T(P_I, Q_I)(\alpha)$. According to Lemma 3.1 and Proposition B.2, we have

$$f(\alpha(t)) = \sum_{i=1}^{m} w_i f_i(\alpha_i(t))$$

with $f_i = T(P_i, Q_i)$ and $\alpha_i(t) = (f_i')^{-1}(t)$. Note that

$$\frac{df}{d\alpha} = \frac{df}{dt}\frac{dt}{d\alpha} = \frac{\sum_{i=1}^{m} w_i f_i'(\alpha_i(t))\alpha_i'(t)}{\sum_{i=1}^{m} w_i \alpha_i'(t)} = -t,$$

where the second equality is a result of the inverse function theorem and the last equality is because $f_i'(\alpha_i(t)) = -t$. Thus, using Lemma 3.1 and Lemma B.2 in [14], we have

$$D_F\left(P_\mathbf{w}\|Q_\mathbf{w}\right) = l_F(T(P_\mathbf{w}, Q_\mathbf{w})) \le l_F(f) = \int_0^1 F\left(\frac{1}{|f'(\alpha)|}\right)|f'(\alpha)|d\alpha$$

$$= \int_0^\infty F\left(\frac{1}{t}\right)\cdot t\cdot\frac{d\alpha}{dt}dt = \sum_{i=1}^{m} w_i \int_0^\infty F\left(\frac{1}{t}\right)\cdot t\cdot\alpha_i'(t)dt.$$

Since

$$\frac{df_i}{d\alpha_i} = \frac{df_i}{dt}\frac{dt}{d\alpha_i} = -t,$$

we have

$$\int_0^\infty F\left(\frac{1}{t}\right)\cdot t\cdot\alpha_i'(t)dt = \int_0^1 F\left(\frac{1}{|f_i'(\alpha_i)|}\right)\cdot|f_i'(\alpha_i)|\cdot d\alpha_i = D_F(P_i\|Q_i)$$

and

$$D_F\left(P_\mathbf{w}\|Q_\mathbf{w}\right) \le \sum_{i=1}^{m} w_i \int_0^1 F\left(\frac{1}{|f_i'(\alpha_i|)}\right)\cdot|f_i'(\alpha_i)|\cdot d\alpha_i = \sum_{i=1}^{m} w_i D_F(P_i\|Q_i).$$

$\square$

Let $H_\gamma(P\|Q) = \int \left[\frac{p(x)}{q(x)} - \gamma\right]_+ dQ(x)$ be the hockey-stick divergence between $P$ and $Q$. Note that the hockey-stick divergence $H_\gamma$ is an $F$-divergence with $F(s) = (s - \gamma)_+$. It holds the following Corollary.

**Corollary B.5** (An application of Proposition 3.3 to the hockey-stick divergence.). *For any $\gamma \ge 1$, we have*

$$H_\gamma(P_\mathbf{w}\|Q_\mathbf{w}) \le \sum_{i=1}^{m} w_i H_\gamma(P_i\|Q_i).$$

Let $R_{\widetilde{\alpha}}(P\|Q)$ be the Rényi divergence of order $\widetilde{\alpha}$ between two distributions $P$ and $Q$. $R_{\widetilde{\alpha}}$ is not an $F$-divergence with convex $F$. However, the scaled exponentiation of Rényi divergence $e^{(\widetilde{\alpha}-1)R_{\widetilde{\alpha}}}$ is known as the power divergence that corresponds to $F_{\widetilde{\alpha}}(s) = \frac{s^{\widetilde{\alpha}} - \widetilde{\alpha}(s-1) - 1}{\widetilde{\alpha}(\widetilde{\alpha}-1)}$. The joint convexity of the scaled exponentiation of Rényi divergence can be derived from Proposition 3.3.

**Corollary B.6** (An application of Proposition 3.3 to the Rényi divergence). *It holds*

$$e^{(\widetilde{\alpha}-1)R_{\widetilde{\alpha}}\left(P_\mathbf{w}\|Q_\mathbf{w}\right)} \le \sum_{i=1}^{m} w_i \int_0^1 |f_i'(x)|^{1-\widetilde{\alpha}}\, dx = \sum_{i=1}^{m} w_i e^{(\widetilde{\alpha}-1)R_{\widetilde{\alpha}}\left(P_i\|Q_i\right)},$$

*for any $\widetilde{\alpha} \ge 1$.*

Corollary B.6 is in line with Lemma 4.1 in [45]. Conversion from a trade-off function to an $F$-divergence is straightforward using Section B in [14]. However, conversion from an $F$-divergence to a trade-off function is highly non-trivial. In fact, the trade-off function is a (global) integral over the whole space while Lemma 3.1 holds pointwisely, which is a local property. This explains why the divergence-based DP is not as informative as $f$-DP since some information is lost due to the integration. Specifically, the following proposition says the length of a trade-off function (which is also a global property) is related to an $F$-divergence.

**Proposition B.7.** *Let $f$ be a trade-off function and let $\mathrm{len}(f)$ be the length of $f$. Then we have*

$$\mathrm{len}(f) = \int_0^1 \sqrt{(1 + f'(x)^2)} = \int_0^1 F\left(\frac{1}{|f_i'(x)|}\right)|f_i'(x)|dx$$

*with $F(y) = \sqrt{1 + y^2}$.*

## C  Technical details for shuffling models in Section 4.1

In this section, we delve into the discussion on shuffling models as introduced in Section 4.1. To specify the distribution of the output of shuffling models, we refer to the work of [23]. Recall that in shuffling models, a dataset $\mathcal{D} \in \mathcal{Z}^n$ is privatized by a local randomizer $\mathcal{A} : \mathcal{Z}^n \to \widetilde{\mathcal{Z}}^n$ that is $\epsilon_0$-DP. Then, a shuffler $\mathcal{A}_{\mathrm{Shuffle}} : \widetilde{\mathcal{Z}}^n \to \widetilde{\mathcal{Z}}^n$ applies a uniformly random permutation to $\mathcal{A}(D)$.

According to Theorem 3.1 in [23], for any two neighboring datasets $\mathcal{D}_0$ and $\mathcal{D}_1$, there is a post-processing procedure $\mathrm{Proc}$ such that $\mathcal{A}_{\mathrm{Shuffle}} \circ \mathcal{A}(\mathcal{D}_0) = \mathrm{Proc}(X)$ and $\mathcal{A}_{\mathrm{Shuffle}} \circ \mathcal{A}(\mathcal{D}_1) = \mathrm{Proc}(Y)$ with $X \sim P$ and $Y \sim Q$. Here $P = (1 - w)P_0 + wQ_0$ and $Q = (1 - w)Q_0 + wP_0$ are two distributions with $(A + 1, C - A) \sim P_0$, $(A, C - A + 1) \sim Q_0$, and $w = 1/(e^{\epsilon_0} + 1)$, where $A \sim \mathrm{Binom}(C, 1/2)$ and $C \sim \mathrm{Binom}(n - 1, 2/(e^{\epsilon_0} + 1))$. As a result of Lemma A.2, it is sufficient to bound $T(P, Q)$.

In the subsequent part of this section, we bound $T(P, Q)$ through the following two steps. First, we bound $T(P_0, Q_0)$ using the joint concavity Lemma 3.1, and the resulting bound is provided by Proposition 4.1. Next, we can establish a bound for the trade-off function $T(P, Q)$ by applying Proposition 6.4.

### C.1  Proof of Proposition 4.1 and Theorem 4.2

In this section, we present the proof of Theorem 4.2. Since the proof relies on Proposition 4.1, we will begin by proving Proposition 4.1.

#### C.1.1  Proof of Proposition 4.1

The upper bound can be derived directly from Lemma 3.1. It is enough to show that the equality in Proposition 4.1 holds.

Let $p_0$ and $q_0$ be the probability density functions of $P_0$ and $Q_0$, respectively. As stated in Lemma A.1, our initial step is to verify the likelihood ratio $p_0/q_0$. For any $(a, b)$ belongs to the support of both $P_0$ and $Q_0$, we have

$$\begin{aligned}
p_0(a, b) = \mathbb{P}\left[A + 1 = a, C - A = b\right] &= \mathbb{P}\left[A_i = a - 1, i - A_i = b \mid C = i\right]\mathbb{P}[C = i] \\
&= \mathbb{P}\left[A_i = a - 1, i = a + b - 1 \mid C = i\right]\mathbb{P}[C = i]
\end{aligned}$$

and

$$q_0(a, b) = \mathbb{P}\left[A = a, C - A + 1 = b\right] = \mathbb{P}\left[A_i = a, i = a + b - 1 \mid C = i\right]\mathbb{P}[C = i].$$

Thus,

$$\frac{p_0(a, b)}{q_0(a, b)} = \frac{\mathbb{P}\left[A_i = a - 1, i = a + b - 1\right]}{\mathbb{P}\left[A_i = a - 1, i = a + b - 1\right]} = \frac{a}{b}.$$

When $p_0(a, b) = 0$ and $q_0(a, b) \neq 0$, we have $a = 0$, $b = C + 1$, and $a/b = 0 = p_0(a, b)/q_0(a, b)$. Similarly, for the case $p_0(a, b) \neq 0$ and $q_0(a, b) = 0$, we have $a = C + 1$, $b = 0$, and $a/b = \infty = p_0(a, b)/q_0(a, b)$. In conclusion, it holds $\frac{p_0(a,b)}{q_0(a,b)} = \frac{a}{b}$.

The corresponding type I error is

$$\alpha_0(t, c) = \mathbb{P}_{A,C}\left[\frac{A+1}{C-A} < t\right] + c\mathbb{P}_{A,C}\left[\frac{A+1}{C-A} = t\right],$$

for any $t \geq 0$ and $c \in [0, 1]$, and the type II error is given by

$$\beta_0(t, c) = \mathbb{P}_{A,C}\left[\frac{A}{C-A+1} > t\right] + (1-c)\mathbb{P}_{A,C}\left[\frac{A}{C-A+1} = t\right].$$

Since the distributions of $A$ and $C$ are discrete, the trade-off function between $\alpha_0$ and $\beta_0$ is piece-wise linear and each knot corresponds to some $t$ with $c = 1$. For simplicity, we define

$$\alpha_0(t) = \alpha_0(t, 1) = \mathbb{P}_{A,C}\left[\frac{A+1}{C-A} \leq t\right]$$

and

$$\beta_0(t) = \beta_0(t, 1) = \mathbb{P}_{A,C}\left[\frac{A}{C-A+1} > t\right].$$

Note that given $C = i$, $(A + 1, C - A) = (A_i + 1, i - A_i)$ with $A_i \sim \text{Binom}(i, 1/2)$. We have $P_0$ is a mixture of $\{(A_i + 1, i - A_i)\}_{i=0}^{n-1}$ and the weights are $\{w_i^0\}_{i=0}^{n-1}$ with $w_i^0 = \mathbb{P}[C = i]$. Using this observation, we rewrite

$$\alpha_0(t) = \mathbb{E}_C\left\{\mathbb{P}\left[\frac{A+1}{C-A} \leq t \,\Big|\, C\right]\right\} = \sum_{i=0}^{n-1} w_i^0\left\{\mathbb{P}\left[\frac{A_i+1}{i-A_i} \leq t\right]\right\} =: \sum_{i=0}^{n-1} w_i^0 \alpha_i^0(t).$$

For each $\alpha_i^0(t)$, it holds

$$\alpha_i^0(t) = \mathbb{P}\left[\frac{A_i+1}{i-A_i} \leq t\right] = F_i\left(i - \frac{i+1}{t+1}\right).$$

Similarly, we can decompose $\beta_0(t) = \sum_{i=0}^{n-1} w_i^0 \beta_i^0(t)$, where

$$\beta_i^0(t) = \mathbb{P}\left[\frac{A_i}{i-A_i+1} > t\right] = 1 - F_i\left(i+1 - \frac{i+1}{t+1}\right).$$

Since

$$\alpha_i^0(t) = F_i\left(i - \frac{i+1}{t+1}\right) = F_i(s_i)$$

with $s_i = F_i^{-1}(\alpha_i^0(t)) \in \mathbb{N}$, we have

$$\beta_i^0(t) = F_i\left(i+1 - \frac{i+1}{t+1}\right) = F_i(s_i + 1) = F_i(F_i^{-1}(\alpha_i^0(t)) + 1),$$

where the second equality is because the support of $F_i$ is $\{0, 1, \cdots, i\}$ and $s_i, s_i + 1 \in \mathbb{N}$. In conclusion, it holds

$$\beta_0(t) = \sum_{i=0}^{n-1} w_i^0 \beta_i^0(t) = \sum_{i=0}^{n-1} w_i^0\left\{F_i(F_i^{-1}(\alpha_i^0(t)) + 1)\right\},$$

which completes the proof.

### C.1.2 Proof of Theorem 4.2

Now we compute the trade-off function $f_{\text{Shuffle}}$ at each knot $\alpha(t)$. Let $f_0 = \mathcal{C}(T(P_0, Q_0))$ be the symmetrization of $T(P_0, Q_0)$ and rewrite $f_{P,Q} = T(P, Q)$.

*Proof of Theorem 4.2.* The proof is a straightforward conclusion from Proposition 6.4. To complete the proof, we still need to show that $\mathcal{C}(f_{P,Q}) = \mathcal{C}(f_{\text{Shuffle}})$ with $f_{\text{Shuffle}} = 2w \cdot \text{Id} + (1 - 2w) \cdot T(P_0, Q_0)$. By the proof of Proposition F.2 in [14], we have $\mathcal{C}(f)(x) = f(x)$ for any $x \leq \bar{x}_f$ and any trade-off function $f$, where $\bar{x}_f$ is such that $\inf\{x \in [0, 1], -1 \in f(x)\}$. Thus, we have $f_0(x) = T(P_0, Q_0)(x)$ for any $x \leq \bar{x}_{f_0}$. Note that $-1 \in \partial f_{P,Q}(x)$ if and only if $-1 \in \partial f_0(x)$. We obtain $\bar{x}_{f_{P,Q}} = \bar{x}_{f_0}$. Moreover, for $x \leq \bar{x}_{f_0}$, it holds

$$f_{P,Q}(x) = 2w\text{Id}(x) + (1 - 2w)f_0(x) = 2w\text{Id}(x) + (1 - 2w)T(P_0, Q_0)(x) = f_{\text{Shuffle}}(x).$$

Using the symmetry of $\mathcal{C}(f_{\text{Shuffle}})$ and $\mathcal{C}(f_{P,Q})$ in Equation (3), we have $\mathcal{C}(f_{\text{Shuffle}}) = \mathcal{C}(f_{P,Q})$. $\square$

## C.2 Proof of Corollary 4.3

According to [14] and the proof of Theorem 4.2, we have $\mathcal{A}_{\text{Shuffle}} \circ \mathcal{A}$ is $(\epsilon, \delta)$-DP with $\delta(\epsilon) = 1 + f_{P,Q}^*(-e^\epsilon)$.

Recall the definition $f^*(y) = \sup_\alpha \{y\alpha - f(\alpha) =: h_y(\alpha)\}$. Then, by the first-order optimality condition, we have $f^*(y) = h_y(\widetilde{\alpha})$ with $\widetilde{\alpha} = \inf\{\alpha \in [0, 1], 0 \in \partial h_y(\alpha)\}$.

For $\alpha(t)$ such that $f_{P,Q}$ is differentiable at $\alpha(t)$, we have $f'_{P,Q}(\alpha(t)) = -2w + (1 - 2w)l(t)$ with

$$l(t) = -\frac{\sum_{i=0}^{n-1} w_i^0 p_i \left(\left\lfloor i + 1 - \frac{i+1}{t+1} \right\rfloor\right)}{\sum_{i=0}^{n-1} w_i^0 p_i \left(\left\lfloor i - \frac{i+1}{t+1} \right\rfloor\right)}.$$

Here $p_i$ is the probability mass function of $A_i$. Thus, $h'_y(\alpha(t)) = y + 2w - (1 - 2w)l(t)$. And $\widetilde{\alpha}(t)$ is then given by $\inf\{\alpha : h'_y(\alpha) \leq 0\}$. Since $\alpha(t)$ is an increasing function of $t$, we obtain

$$\widetilde{\alpha}(t) = \alpha(t_y), \qquad \text{with } t_y = \inf\{t : y + 2w - (1 - 2w)l(t) \leq 0\}$$

and

$$f_{\text{Shuffle}}^*(y) = h_y(\alpha(t_y)) = y\alpha(t_y) - f_{P,Q}(\alpha(t_y)) = y\alpha(t_y) - \beta(t_y).$$

We end the proof by taking $y = -e^\epsilon$.

# D   Omited details of Section 5

## D.1   Proof of Theorem 5.1

According to a continuous version of Lemma 3.1 , that is given by Equation (4) and Equation (5), $T(X, Y)$ in Theorem 5.1 is lower bounded by the trade-off function $T(P_I, Q_I)$ with $I \sim \mathcal{N}(0, 1)$, $P_I = \mathcal{N}(0, 1)$ and $Q_I = \mathcal{N}(\mu_I, 1)$. For this example, we have $p_I(x) = e^{-x^2/2}$ and $q_I(x) = e^{-(x-\mu_I)^2/2}$. Then the type I error is

$$\alpha(t) = \int_{-\infty}^{\infty} \mathbb{P}_{X \sim \mathcal{N}(0,1)} \left[-\mu_w X + \frac{\mu_w^2}{2} \leq t\right] e^{-\frac{w^2}{2}} dw$$

$$= \int_{\mu_w \leq 0} \Phi(t_w) e^{-\frac{w^2}{2}} dw + \int_{\mu_w > 0} \Phi(-t_w) e^{-\frac{w^2}{2}} dw$$

with $t_w = -\frac{t}{\mu_w} + \frac{\mu_w}{2}$. Similarly, the type II error is

$$\beta(t) = \int_{-\infty}^{\infty} \mathbb{P}_{X \sim \mathcal{N}(\mu_w, 1)} \left[-\mu_w X + \frac{\mu_w^2}{2} > t\right] e^{-\frac{w^2}{2}} dw$$

$$= \int_{\mu_w \leq 0} [\Phi(-t_w + \mu_w)] e^{-\frac{w^2}{2}} dw + \int_{\mu_w > 0} [\Phi(t_w - \mu_w)] e^{-\frac{w^2}{2}} dw,$$

which completes the proof of Theorem 5.1.

## D.2   Examples for different loss functions

Recall the noiseless linear model with $\mathcal{D}_0 = \{(x_i, y_i)\}_{i=1}^n$ with $y_i = ax_i$ and $x_i^2 = 1$ for some constant $a$ and we defined $\mathcal{D}_1$ by removing an arbitrary element in $\mathcal{D}_0$.

**Example D.1** (Least-squares loss without gradient clipping)**.** *For linear least squares regression with* $\ell(\theta, \mathcal{D}) = \sum_{i=1}^n (y_i - \theta x_i)^2$ *, we have* $g(\theta, \mathcal{D}) = \sum_{i=1}^n (y_i - \theta x_i)(-x_i)$ *and* $\mu_I = a - I$.

In Example D.1, the gradient is unbounded due to an unbounded initializtion, and so is its sensitivity. In this example, the dominate pair for $\theta(\mathcal{D}_0)$ and $\theta(\mathcal{D}_1)$ is $(\mathcal{N}(0, \sigma^2), \mathcal{N}(0, \sigma^2) + a - I)$. Note that $\mathcal{N}(0, \sigma^2) + a - I$ is a Gaussian distribution with mean $a$ and variance $1 + \sigma^2$. Thus, under the framework of RDP, the goal is to bound the Rényi divergence between two Gaussian distributions with different variances, which is unbounded for $\widetilde{\alpha}$ large enough.

**Example D.2** (Least squares loss with gradient clipping). *Consider a linear least squares regression problem in Example D.1. For DP-GD with gradient clipping, we have*

$$\mu_I = \begin{cases} a - I, & |a - I| \leq c, \\ c, & a - I \geq c, \\ -c, & a - I \leq -c. \end{cases}$$

**Example D.3** (Logistic loss). *For the logistic loss, we have*

$$\mu_I^{\max} = \sup_{x,y} \left| \frac{e^{-I \cdot yx}}{1 + e^{-I \cdot yx}} \right|$$

*as the gradient of the logistic loss is the softmax function. $\mu_I^{\max}$ is bounded when $|xy| \leq M$ for some $M > 0$. Furthermore, extending the logistic loss to other strongly convex losses is straightforward, given that the key feature is the gradient being a monotone function of $I \cdot yx$.*

## E   Technical details of Lemma 6.3 and corresponding conclusions

In this section, we discuss the omitted details of Section 6 including the proofs of the advanced joint concavity (Lemma 6.3).

### E.1   Proof of Lemma 6.2, Lemma 6.3, and corresponding results

In this section, we establish the proof of Lemma 6.2 and Lemma 6.3. Before delving into the proof, we revisit Proposition 6.4 that directly stem from the application of Lemma 6.2 and Lemma 6.3. The proof of Proposition 6.4 is included as part of the proof of Theorem 4.2 in Section E.1.1. Similar to the proof of Proposition 6.4, for $P = (1 - w)P_0 + wP_1$ and $Q = (1 - w)P_0 + wQ_1$ that appear in the analysis sub-sampling [4], we have the following proposition.

**Proposition E.1.** *For $P = (1 - w)P_0 + wP_1$ and $Q = (1 - w)P_0 + wQ_1$, we have*

$$T(P, Q) \geq \mathcal{C}\left((1 - w)\mathrm{Id} + wT(P_1, Q_1)\right).$$

*Proof of Lemma 6.2.* We first invoke an important equality from [4].

$$H_\alpha(P\|Q) := \sup_E \{P(E) - \alpha Q(E)\} = \int [p(z) - \alpha q(z)]_+ \, dz. \tag{8}$$

According to [5], we have $\mathcal{A}$ is $(\epsilon, \delta)$-differentially private if and only if $H_{e^\epsilon}(\mathcal{A}(D_0)\|\mathcal{A}(D_1)) \leq \delta$ for every neighboring $D_0$ and $D_1$. We now recall the following two equations which are constraints on $\epsilon_0, \epsilon_1, \gamma$, and $\eta$:

$$e^{\epsilon'} = (1 - w)e^{\epsilon_0} + we^{\epsilon_1} \tag{9}$$

and

$$we^{\epsilon'} = (1 - w)\gamma e^{\epsilon_0} + w\eta e^{\epsilon_1}. \tag{10}$$

It is evident from Equations (9) and (10) that

$$\exp(\epsilon_0)(1 - w)\gamma + \exp(\epsilon_1)w\eta = \exp(\epsilon')w,$$

and

$$\exp(\epsilon_0)(1 - w)(1 - \gamma) + \exp(\epsilon_1)w(1 - \eta) = \exp(\epsilon')(1 - w).$$

Thus, we have

$$(1 - w)P_1 + wP_2 - e^{\epsilon'}\left((1 - w)Q_1 + wQ_2\right)$$
$$= (1 - w)\left(P_1 - e^{\epsilon_0}(1 - \gamma)Q_1 - e^{\epsilon_0}\gamma Q_2\right) + w\left(P_2 - e^{\epsilon_1}(1 - \eta)Q_1 - e^{\epsilon_1}\eta Q_2\right).$$

This completes the proof of proposition by equation (8). □

*Proof of Lemma 6.3.* According to Lemma 6.2 and Proposition 2.12 in [14], we aim to find a trade-off function $F$ such that

$$F^*(-e^{\epsilon'}) = (1-w)(1-\gamma)f_{0,0}^*(-e^{\epsilon_0}) + w(1-\eta)f_{1,0}^*(-e^{\epsilon_1})$$
$$+ (1-w)\gamma f_{0,1}^*(-e^{\epsilon_0}) + w\eta f_{1,1}^*(-e^{\epsilon_1}), \tag{11}$$

where $\epsilon_0, \epsilon_1, \gamma, \eta$ satisfy equations (9) and (10). Let $y = -e^{\epsilon'} < -1$ and, for fixed $\gamma < w < \eta$, Equations (9) and (10) imply that

$$-e^{\epsilon_0} = \frac{y(\eta-w)}{(1-w)(\eta-\gamma)}, \quad \text{and} \quad -e^{\epsilon_1} = \frac{(w-\gamma)y}{w(\eta-\gamma)}.$$

Therefore, $F(x)$ is given by the double conjugate:

$$F(x) = \sup_y xy - F^*(y)$$

$$= \sup_y xy - (1-w)(1-\gamma)f_{0,0}^*\left(\frac{y(\eta-w)}{(1-w)(\eta-\gamma)}\right) - w(1-\eta)f_{1,0}^*\left(\frac{(w-\gamma)y}{w(\eta-\gamma)}\right)$$

$$- (1-w)\gamma f_{0,1}^*\left(\frac{y(\eta-w)}{(1-w)(\eta-\gamma)}\right) - w\eta f_{1,1}^*\left(\frac{(w-\gamma)y}{w(\eta-\gamma)}\right).$$

For $i = 0, 1$, define

$$F_{0,i}(x) = f_{0,i}\left(\frac{x(1-w)(\eta-\gamma)}{(\eta-w)}\right), \quad \text{and} \quad F_{1,i}(x) := f_{1,i}\left(\frac{xw(\eta-\gamma)}{(w-\gamma)}\right).$$

Thus, we have

$$F_{0,i}^*(y) = f_{0,i}^*\left(\frac{y(\eta-w)}{(1-w)(\eta-\gamma)}\right), \quad \text{and} \quad F_{1,i}^*(y) = f_{1,i}^*\left(\frac{(w-\gamma)y}{w(\eta-\gamma)}\right).$$

Therefore, it holds

$$F(x) = \sup_y \left\{ xy - (1-w)(1-\gamma)F_{0,0}^*(y) - w(1-\eta)F_{1,0}^*(y) - (1-w)\gamma F_{0,1}^*(y) - w\eta F_{1,1}^*(y) \right\}$$

$$= \left( (1-w)(1-\gamma)F_{0,0}^* + w(1-\eta)F_{1,0}^* + (1-w)\gamma F_{0,1}^* + w\eta F_{1,1}^* \right)^*(x),$$

for all possible $\gamma < w < \eta$. Similar results for $\eta < w < \gamma$ can be obtained by symmetry.

When $\eta = \gamma = w$, we would like to show

$$T((1-w)P_0 + wP_1, (1-w)Q_0 + wQ_1)$$
$$\geq (1-w)T(P_0\|(1-w)Q_0 + wQ_1) + wT(P_1\|(1-w)Q_0 + wQ_1).$$

Rewrite $F_1 = T(P_0\|(1-w)Q_0 + wQ_1)$ and $F_2 = T(P_1\|(1-w)Q_0 + wQ_1)$. Lemma 6.2 implies that

$$F^*(-e^{\epsilon'}) = \inf_{e^{\epsilon_0}, e^{\epsilon_1}} \left\{ (1-w)F_1^*(-e^{\epsilon_0}) + wF_2^*(-e^{\epsilon_1}) | (1-w)e^{\epsilon_0} + we^{\epsilon_1} = e^{\epsilon'} \right\},$$

where the constraint $(1-w)e^{\epsilon_0} + we^{\epsilon_1} = e^{\epsilon'}$ comes from equation (9) and (10). Thus, for any $x \in [0, 1]$, we have

$$F(x) = \sup_{-e^{\epsilon'}, \epsilon'>0} -xe^{\epsilon'} - F^*(-e^{\epsilon'})$$

$$= \sup_{e^{\epsilon'}, \epsilon'>0} -xe^{\epsilon'} - \inf_{e^{\epsilon_0}, e^{\epsilon_1}} \left\{ (1-w)F_1^*(-e^{\epsilon_0}) + wF_2^*(-e^{\epsilon_1}) | (1-w)e^{\epsilon_0} + we^{\epsilon_1} = e^{\epsilon'} \right\}$$

$$= \sup_{e^{\epsilon'}} \sup_{e^{\epsilon_0}, e^{\epsilon_1}:(1-w)e^{\epsilon_0}+we^{\epsilon_1}=e^{\epsilon'}} -xe^{\epsilon'} - \left\{ (1-w)F_1^*(-e^{\epsilon_0}) + wF_2^*(-e^{\epsilon_1}) \right\}.$$

According to the properties of infimal convolution in convex analysis (cf., Exercise 12 of Chapter 3.3 in Page 57 of [8]), we get

$$F(x) = \sup_{e^{\epsilon_0}, e^{\epsilon_1}} -(1-w)xe^{\epsilon_0} - wxe^{\epsilon_1} - (1-w)F_1^*(-e^{\epsilon_0}) - wF_2^*(-e^{\epsilon_1})$$

$$= \sup_{e^{\epsilon_0}} -(1-w)xe^{\epsilon_0} - (1-w)F_1^*(-e^{\epsilon_0}) + \sup_{e^{\epsilon_1}} -wxe^{\epsilon_1} - wF_2^*(-e^{\epsilon_1})$$

$$= (1-w)F_1(x)^{**} + wF_2(x)^{**} = (1-w)F_1(x) + wF_2(x).$$

This completes the proof of this corollary. $\qquad\square$

### E.1.1 Proof of Proposition 6.4

Let $f_0 = \mathcal{C}(T(P_0, Q_0))$ be the symmetrization of $T(P_0, Q_0)$ and rewrite $f_{P,Q} = T(P, Q)$.

*Proof of Proposition 6.4.* Since $P = (1-w)P_0 + wQ_0$ and $Q = (1-w)Q_0 + wP_0$, according to Theorem 2 in [4], we have

$$
\begin{aligned}
H_{e^{\epsilon'}}(P\|Q) &= H_{e^{\epsilon'}}((1-w)P_0 + wQ_0\|(1-w)Q_0 + wP_0) \\
&= H_{e^{\epsilon'}}\left(2w \cdot \frac{P_0 + Q_0}{2} + (1-2w)P_0 \middle\| 2w \cdot \frac{P_0 + Q_0}{2} + (1-2w)Q_0\right) \\
&\leq (1-2w)H_{e^{\epsilon}}\left(P_0 \middle\| (1-\eta) \cdot \frac{P_0 + Q_0}{2} + \eta Q_0\right) \\
&\leq (1-2w)\left(\frac{1}{2} + \frac{\eta}{2}\right) H_{e^{\epsilon}}(P_0\|Q_0)
\end{aligned}
$$

with $e^{\epsilon'} = (1-2w)e^{\epsilon} + 2w$ and $\eta = e^{\epsilon'}/e^{\epsilon}$. It is clear from the definition that $\eta \leq 1$. Therefore,

$$
H_{e^{\epsilon'}}(P\|Q) \leq (1-2w)H_{e^{\epsilon}}(P_0\|Q_0).
$$

Let $y = -e^{\epsilon'}$. Proposition 2.12 in [14] implies that

$$
1 + f_{P,Q}^*(y) \leq (1-2w)\left(1 + f_0^*\left(\frac{y + 2w}{1 - 2w}\right)\right)
$$

Therefore,

$$
\begin{aligned}
f_{P,Q}(x) &= \sup_y xy - f_{P,Q}^*(y) \\
&\geq \left(-2w_0 + (1-2w_0)f_0^*\left(\frac{y + 2w}{1 - 2w}\right)\right)^*
\end{aligned}
$$

By properties of convex conjugate, we have

$$
\begin{aligned}
f_{P,Q}(x) &\geq 2w(1-x) + (1-2w)f_0^{**}(x) \\
&= 2w(1-x) + (1-2w)f_0(x)
\end{aligned}
$$

According to Proposition F.2 in [14], the shuffling model is $\mathcal{C}(f_{P,Q})$-DP. $\qquad\square$

## F Tightness of Lemma 3.1

As we see from Proposition 4.1, Lemma 3.1 holds with equality. However, in general, Lemma 3.1 is not tight (cf., Figure 3b). From the technical proof of Proposition 4.1, we obtain that $\frac{p_{\mathbf{w}}}{q_{\mathbf{w}}} = \sum_{i=1}^m w_i p_i/q_i$, which motivates us to derive Proposition 6.1.

### F.1 Proof of Proposition 6.1

By Theorem 2.10 in [14], we know that for distributions $P_{\mathbf{w}}, Q_{\mathbf{w}}$ and $P_I, Q_I$, it holds

$$
T(P_{\mathbf{w}}, Q_{\mathbf{w}}) \geq T(P_I, Q_I) \quad \text{iff} \quad (P_{\mathbf{w}}, Q_{\mathbf{w}}) \succeq_{Blackwell} (P_I, Q_I).
$$

We define the Blackwell order as in, for example, [7, 14, 34]. Precisely, if there are probability distributions $P$ and $Q$ on $Y$, as well as probability distributions $P'$ and $Q'$ on $Z$, and a randomized algorithm $\text{Proc} : Y \mapsto Z$ such that $\text{Proc}(P) = P', \text{Proc}(Q) = Q'$, then we write $(P, Q) \succeq_{Blackwell} (P', Q')$.

Let $F_0$ be the cumulative distribution function of the log-likelihood ratio $\log \frac{dP_{\mathbf{w}}}{dQ_{\mathbf{w}}}(X)$ for $X \sim P_{\mathbf{w}}$. $G_0$ is defined analogously by replacing $P_{\mathbf{w}}$ and $Q_{\mathbf{w}}$ with $P_I$ and $Q_I$, respectively. Furthermore, we define the perfect log-likelihood function $\tilde{F}_1(x)$ and $\tilde{G}_1(x)$ to satisfy the following:

$$
\tilde{F}_1(x) = Q_{\mathbf{w}}\left(\log\left(\frac{dQ_{\mathbf{w}}}{dP_{\mathbf{w}}}\right) - E \leq x\right),
$$

and

$$\tilde{G}_1(x) = Q_I\left(\log\left(\frac{dQ_I}{dP_I}\right) - E \le x\right),$$

where $E$ is a random variable such that, under $Q_\mathbf{w}$, $E$ is independent of $\log\frac{dQ_\mathbf{w}}{dP_\mathbf{w}}$ and is distributed according to an exponential distribution with support $\mathbb{R}_+$ and cumulative distribution function $1 - e^{-x}$ for all $x \ge 0$. By Theorem 3 in [32], we know

$$\tilde{F}_1(x) \ge \tilde{G}_1(x), \qquad \text{for all } x \in \mathbb{R},$$

if and only if

$$(P_\mathbf{w}, Q_\mathbf{w}) \succeq_{Blackwell} (P_I, Q_I).$$

Therefore, equality in Lemma 3.1 holds if and only if $\tilde{F}_1(x) = \tilde{G}_1(x)$ for all $x \in \mathbb{R}$. The following equations (12) and (13) is appear in the proof of Lemma 1 in [32]. For the sake of thoroughness, we will include a summary of the proof later in this section for reference. We have

$$\tilde{F}_1(x) = \int_{-x}^{\infty} F_0(v)e^{-v}dv, \tag{12}$$

and

$$\tilde{G}_1(x) = \int_{-x}^{\infty} G_0(v)e^{-v}dv. \tag{13}$$

Since $F_0$ and $G_0$ are continuous, equality holds for all $x$ if and only if $F_0(v)e^{-v} = G_0(v)e^{-v}$ by fundamental theorem of calculus. We conclude that equality in Lemma 3.1 holds if and only if

$$\frac{w_1p_1 + w_2p_2}{w_1q_1 + w_2q_2} \overset{\mathbb{P}}{=} w_1\frac{p_1}{q_1} + w_2\frac{p_2}{q_2}$$

with respect to $P_\mathbf{w}$.

*Proof of Equation* (12) *and* (13). We define

$$F_1(v) = Q_\mathbf{w}\left(\log\frac{dQ_\mathbf{w}}{dP_\mathbf{w}} \le v\right)$$

and $\tilde{F}_1$ to be the convolution of the distribution $F_1$ with the distribution of $-E$, and thus can be written as

$$\tilde{F}_1(x) = \int_{\mathbb{R}} Q_\mathbf{w}(-E \le x - u)dF_1(u)$$

$$= F_1(x) + e^x \int_x^{\infty} e^{-u}dF_1(u)$$

$$= \int_{-\infty}^{x} dF_1(u) + e^x \int_x^{\infty} e^{-u}dF_1(u).$$

Moreover, we substitute that $dF_1(u) = -e^u dF_0(-u)$ into equations above. Then, it holds

$$\tilde{F}_1(x) = \int_{-\infty}^{x} -e^u dF_0(-u) + e^x \int_x^{\infty} -dF_0(-u)$$

$$= \int_{-x}^{\infty} e^{-u}dF_0(u) + e^x \int_{-\infty}^{-x} dF_0(u)$$

$$= \int_{-x}^{\infty} -e^u dF_0(u) + e^x F_0(-x).$$

We conclude equation (12) via integral by part. Equation (13) can be proved similarly. $\qquad\square$

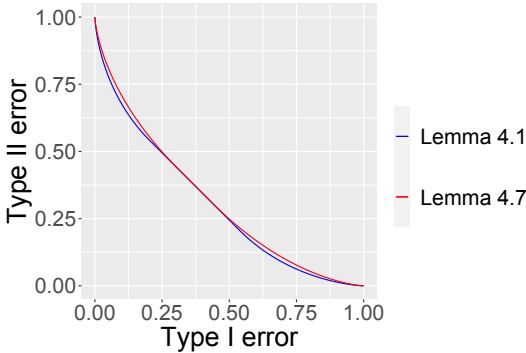

Figure 4: Example F.3 with $\mu = 1$ and $w = 1/3$.

## F.2 Other examples where Lemma 3.1 holds with equality

According to Proposition 6.1, we may find other examples in which Lemma 3.1 holds with equality. Straightforward examples are that the support of $w_1 P_1 + w_2 P_2$ and the support of $w_1 Q_1 + w_2 Q_2$ are disjoint sets.

**Example F.1.** *Consider* $P_1 = \mathrm{Unif}([0,1])$, $Q_1 = \mathrm{Unif}([2,3])$, $P_2 = \mathrm{Unif}([-1,0])$, *and* $Q_2 = \mathrm{Unif}([3,4])$. *It is easy to verify that the condition in Proposition 6.1 holds. In fact, we have*

$$\frac{P_{\mathbf{w}}}{Q_{\mathbf{w}}}(X) = \frac{P_1}{Q_1}(X) = \frac{P_2}{Q_2}(X) = \infty$$

*as the support of* $Q_1$ *and* $Q_2$ *are disjoint with* $[-1,1]$.

**Example F.2.** *Another example where the equality holds in Lemma 3.1 is that* $P_1$ *and* $Q_1$ *are two probability distributions supported on the* $x$-axis, *and* $P_2$ *and* $Q_2$ *are two probability distributions supported on the* $y$-axis. *If the point mass at* $0$ *are all* $0$, *then one can show that* $P_i$ *and* $Q_i$ *for* $i \leq 2$ *satisfy the equality condition in Lemma 3.1. To see this, let* $X = (X_1, X_2) \in \mathbb{R}^2$ *be a random variable with distribution* $w_1 P_1 + w_2 P_2$. *Therefore,* $X$ *is supported on the axes. For any* $t \geq 0$, *the right hand side of Proposition 6.1 becomes*

$$\mathbb{P}\left(\frac{w_1 p_1 + w_2 p_2}{w_1 q_1 + w_2 q_2}(X) \leq t\right)$$
$$= w_1 \mathbb{P}\left(\frac{w_1 p_1 + w_2 p_2}{w_1 q_1 + w_2 q_2}(X) \leq t \,\Big|\, X \sim P_1\right) + w_2 \mathbb{P}\left(\frac{w_1 p_1 + w_2 p_2}{w_1 q_1 + w_2 q_2}(X) \leq t \,\Big|\, X \sim P_2\right)$$
$$= w_1 \mathbb{P}\left(\frac{p_1}{q_1}(X) \leq t \,\Big|\, X \sim P_1\right) + w_2 \mathbb{P}\left(\frac{p_2}{q_2}(X) \leq t \,\Big|\, X \sim P_2\right),$$

*which is exactly the right-hand side of Proposition 6.1.*

## F.3 Comparisons between Lemma 3.1 and Lemma 6.3

In this section, we compare Lemma 3.1 with Lemma 6.3 using other examples besides that given by Figure 3. The first example is a simple case appears in sub-sampled Gaussian mechanisms.

**Example F.3.** *Let* $P_0 = \mathcal{N}(0,1)$ *and* $Q_0 = \mathcal{N}(\mu,1)$. *Then, using Lemma 6.3, we have* $T(P_0, w P_0 + (1-w)Q_0) \geq \mathcal{C}(w \cdot \mathrm{Id} + (1-w)T(P_0, Q_0))$. *Lemma 3.1 leads to the following lower bound*

$$T(P_0, w P_0 + (1-w)Q_0)(\alpha) \geq \begin{cases} (1-w)\Phi(\Phi^{-1}\left(\frac{1-\alpha}{1-w}\right) - \mu), & \alpha > 1 - (1-w)\Phi(\frac{\mu}{2}), \\ w + (1-w)\Phi\left(\Phi^{-1}\left(1 - \frac{\alpha}{1-w}\right) - \mu\right), & \alpha < (1-w)(1 - \Phi(\frac{\mu}{2})), \\ linear, & otherwise. \end{cases}$$

*As we see from Figure 4, Lemma 6.3 leads to a slightly tighter bound.*

Another example is an extreme case where the mixture distributions are not distinguishable at all.

**Example F.4.** *Consider the case $(\frac{1}{2}P + \frac{1}{2}Q)v.s.(\frac{1}{2}Q + \frac{1}{2}P)$ where two distributions are not distinguishable at all. We have $T\left((\frac{1}{2}P + \frac{1}{2}Q), (\frac{1}{2}Q + \frac{1}{2}P)\right) = \mathrm{Id}$ which is a special case of Proposition 6.4. The advanced joint concavity Proposition 6.4 leads to a sharp lower bound $\mathrm{Id}$. However, it is obvious that Lemma 3.1 implies a loose bound when $P \neq Q$.*