# OpenReview forum: "Unified Enhancement of Privacy Bounds for Mixture Mechanisms via $f$-Differential Privacy"
_NeurIPS.cc/2023/Conference — NeurIPS 2023 poster_

### Official Review · Reviewer_xB9B · 2023-07-03

**Soundness:** 3 good
**Presentation:** 3 good
**Contribution:** 3 good
**Rating:** 7
**Confidence:** 4

**Summary:**

This paper introduces a framework for analyzing mixture distributions via $f$-DP and its tradeoff function. Additionally, the paper leverages this framework for improving bounds for shuffling mechanisms in DP, and the same framework to prove a statement about privacy of a single-step of gradient descent from random initialization.

**Strengths:**

This paper addresses a fundamental problem in DP: computing effective estimates on additive mixtures of random variables. Methods of analysis for these mixtures are, to date, reasonably ad-hoc and certainly not known to be tight. This paper presents a clean unified approach, making connections to more abstract and clean mathematically-posed problems (IMO, this is the manner in which DP should imagine evolving itself, away from its CS + statistics formulations, which are often focused on data structures and properties of particular random variables). Not only does this paper formulate a nice presentation, but  to my knowledge it improves existing statements through this framework. Clear accept.

**Weaknesses:**


Writing some nits no presentation that came up from a stream of consciousness read below. Not necessarily all weaknesses, just comments.

* I would prefer a slight reordering of the content. AFAICT, the main logical flow is: Lemma 4.1 -> Prop 4.3 -> Thm 3.1, with the rest of the stuff more or less as add-ons (though FWIW i have not yet read the section on advanced concavity). I would really prefer this logical flow to be more clear. Ideally, for me, we would present the proofs straight through, beginning with the lemmas. However, I get that this is CS and we don't really do that here. I think a reasonable option is to sketch the proof in the main body, or discuss that the major results seem to follow primarily from the fairly simple Lemma 4.1, plus some calculations, given the reduction introduced by [20]. This logical flow is simple and quite nice, and IMO it should be highlighted on its own terms.

* Editorially, I would suggest pushing the DP-GD stuff into an appendix. I think it confuses the message, and it's not clear either how extensible it is (presumably it breaks after one step due to the injected dependence? Though I could imagine conditioning on stuff and invoking nested mixtures, this may be prohibitively difficult since IIUC Gaussian-ness will go away).

* Confusion on 'the outputs of two neighboring datasets are post-processing of r.v.s $X$ and $Y$' (beginning of section 4). Going back to the reference, it seems like the technical meat there is effectively a good reduction to scalar-valued random variables (if  I read correctly, the content of their Thm 3.1)--presumably that is what is invoked here? But it reads as if the development were restricted to scalar-valued mechanisms (which I assume it is not).

* Setup to 4.1: the way that $X$, $Y$, and $I$ are coupled is unstated. Meaning: in the setup here, it is possible that $I$ is independent of $X$ and $Y$ (in which case $X|I$ and $Y|I$ would just be $X$ and $Y$). I guess the coupling is something like, e.g. $X$ is coupled with $I$ such that $X|\{I=i\} \sim p_i$? But if so, I don't see this stated anywhere before the proof of Lemma 4.1.

* I know this is not how CS papers are generally written, but (being an old mathematician myself) I would really personally prefer to read this paper straight through and without applications, building up to an improved analysis (which I assume is Thm 3.1), then closing with discussion of relation to prior results and tightness.

* Speaking of relation to prior results, and putting my CS hat back on: this could really use some deeper dives / commentary. I would recommend cutting the DP-GD stuff (or pushing to an appendix) and extending the discussion of relationship to prior bounds. For example: there is a table showing $\delta$s for varying $\epsilon$s at a fixed $\epsilon_0$. What happens when we vary this $\epsilon_0$ over a wide range--are the results here _always_ better (and qualitatively similar)? Or are there some regimes in which previous analyses were tighter?

* Following the citations of [20] leads me to https://arxiv.org/pdf/2304.05007.pdf. Glancing through this paper, it seems to me that one major difference with the present work is that this work claims improved amplification relative to [20] _independent_ of mechanism structure, whereas that work improves their bounds in the case of some particular mechanisms (and their worst-case bounds, for general mechanisms, are identical). Is this correct? Can you speak to this relationship?

* One confusing thing on a read-through: it is difficult to find the proof of proposition 4.3. This proof can be found in the 'technical details for section 3.1', which is not where a reader like me would look right away (given, in particular, that there is an appendix section on proofs for section 4).

**Questions:**

Simply a summary of some of my stream-of-consciousness comments above:

* Could you comment more extensively on relation to prior results? Is the analysis here _always_ stronger?
* Similarly, could you comment on the relation to https://arxiv.org/pdf/2304.05007.pdf? I listed my understanding of the way these relate above, but looking for some clarity here.

**Limitations:**

Tightness or lack thereof of particular results was discussed. Negative social impacts not immediately applicable.

---

> ### Author Rebuttal · Authors · 2023-08-07
>
> Thank you very much for your extremely positive feedback and your valuable comments. I have organized the responses to the weaknesses and questions as follows, corresponding to the 8 items in weaknesses. The reference number provided here aligns with the supplementary material, which slightly differs from the 9-page paper.
>
> 1. We have taken your advice into account concerning the structure of our paper. Initially, we highlighted the theoretical part as you metioned, but we finally moved it behind the applications section in the submitted version. This adjustment is aimed at enhancing the paper's accessibility to a wider audience, particularly NeurIPS readers, given the statistical nature of the theory. After careful consideration, we have opted to retain the current structure with certain modifications. We have clarified the proof of Theorem 3.1, which relies on both Proposition 4.3 (implied by Lemma 4.1) and Proposition 4.8 (a result of Lemma 4.7). In the submitted version, a proof sketch is contained at the beginning of Section 4 implicitly which has also been highlightened in the revised version. Moreover, a detailed citation and some discussions regarding the post-processing in [23] has also been provided at Section 4.
>
>
> 2. We sincerely value your feedback regarding the DP-GD section (Section 3.2). In the submitted version, we show that it can be applied to noiseless linear models with least squares loss, which are also used in practice (cf., https://arxiv.org/abs/2006.08212). We acknowledge the need to address more general models and the difficulty to clarify the sensitivity $\mu_I$ with given $I$ for general models and loss functions. While we have decided to retain this section, we included more discussions on the extensions and restrictions in the revised version.
>
>
> 3. Both $X$ and $Y$ are 2-dimensional vector-valued random vectors, representing mixtures of binomial distributions.  As you indicated, the primary contribution of [23] is that they propose a promising post-processing procedure to convert the multi-dimensional shuffled model to 2-dimensional random vectors, which makes the likelihood ratio easy to verify.
>
> 4. We have addressed the potentially confusing notation $(X|I, I)$ and introduced further clarification: $(X|I, I)$ means we observe two random variables, $X|I$ and $I$, where $X$ follows a mixture distribution $\sum_{i=1}^m w_i p_i$, and $I$ represents the information of indices with $\mathbb{P}[I=i] = w_i$. $X|I$ denotes $X|I=i \sim p_i$. As releasing the indices exposes more information, the couple $((X|I, I), (Y|I, I))$ is more separable than $(X, Y)$ without knowing the indices, which motivates Lemma 4.1.
>
> 5. We have introduced an explanatory paragraph regarding tightness immediately following Theorem 3.1: The bound in Theorem 3.1 is near-optimal in theory, and it represents a significant enhancement compared to [23], as demonstrated in Table 1 and Figure 1. In fact, the proof of Theorem 3.1 is based on a post-processing procedure in [23], joint concavity (Proposition 4.3), and advanced joint concavity (Proposition 4.8). The post-processing procedure is sharp for specific mechanisms, such as the randomized response mechanism, as shown by Theorem 5.2 and Theorem 5.3 in [23]. Proposition 4.3 is also sharp, as demonstrated in Section 4. Based on the proof of Proposition 4.8 in Section B.1.2, the advanced joint concavity used in this example is almost the tightest closed-form bound that can be derived. Compared to [23], the main advantage of using $f$-DP (Proposition 4.3) is that we avoid the use of Hoeffding's inequality, which is adopted in [22, 23] and leads to loose bounds, to bound the mixture of binomial distributions. Moreover, Theorem 3.2 in [23] holds with an assumption on the local privacy budget $\epsilon_0$, which is removed by using $f$-DP in our paper.
>
>    Consequently, our bound is always better than the theoretical bound in Theorem 3.2 in [23] as Proposition 4.3 is tight and an $(\epsilon,\delta)$-DP version of Proposition 4.8 was also adopted in [23].
>
> 6. In line with Item 5, we have discussed the tightness of the shuffling models in the revised version. Additionally, we have included Figure 6 in the attached pdf about varying $\epsilon_0$ for your reference.
>
>
>     Considering the DP-GD stuff, we discussed the application and extensions from noiseless linear models as stated in Item 2. Since the revised paper does not exceed the pages required by the camera-ready version, we decided to keep this content.
>
> 7.  Thank you for introducing the concurrent work https://arxiv.org/pdf/2304.05007.pdf. After reading through their paper, as you mentioned, one difference is that they analyze specific local randomizers satisfying certain assumptions. Their tightness is to design a tighter post-processing procedure (in comparison to [23]) to reduce the coupled distributions to the mixture of binomial noise for specific local randomizers. Their analysis of the mixture distribution similarly rests on Hoeffding's inequality, which, as discussed in Item 5, is refined by our Proposition 4.3. Combining their novel post-processing with our tight analysis of mixture distributions could potentially yield tighter bounds, an exciting avenue to explore.
>
> 8. We have updated the title of Section B.1 to "Proofs of Proposition 4.3, Proposition 4.8, and Theorem 3.1." Additionally, we have divided Section B.1.2 into two separate sections, one focusing on the proof of Proposition 4.8 and the other on the proof of Theorem 3.1. This reorganization aims to improve clarity and facilitate better understanding for the readers.

---

> > ### Comment · Reviewer_xB9B · 2023-08-13
> >
> > Thank you for your detailed rebuttal. This all makes sense. I will keep my score of accept.

---

> > > ### Author Response · Authors · 2023-08-15
> > >
> > > We are grateful for your valuable comments and comprehensive suggestions once more. We will thoroughly reconsider your insights regarding the paper's structure until we finalized the camera-ready version.

---

### Official Review · Reviewer_wgR7 · 2023-07-04

**Soundness:** 4 excellent
**Presentation:** 3 good
**Contribution:** 4 excellent
**Rating:** 8
**Confidence:** 5

**Summary:**

One of the main challenges the differential privacy frame work is facing these days, is the gap between the variety of randomization techniques applied in the machine learning community for various reasons, and our limited capability to prove the privacy amplifications they entail. Among primary examples we can mention random initialization and shuffling techniques.

The authors of this paper leverage the known connection between DP and hypothesis testing, specifically relaying on the newly presented notion of f-DP, to transition the analysis of the effect of these mechanism from the domain of privacy loss distribution to the domain of hypothesis testing, where they prove several key results. Combining these results with the known implications between f-DP and DP, they provide tighter bounds for privacy amplification by shuffling and new results for privacy amplification by random initialization (for the limited setting of one gradient step).

**Strengths:**

This paper partially fills a long time gap in our understanding of the way privacy is improved by introducing various random steps to the learning algorithm. Unlike most previous works, the authors use the newly presented f-DP definition and the known implications between it and the DP definition, to analyze the effect of these randomization techniques on the trade-off function.

The contribution of this paper extends beyond its results, as the proof techniques presented in it might be used for the analysis of other potential amplification techniques as well.

**Weaknesses:**

At first glance, I got the impression that the results of this paper aim to provide guarantees using a separate notion of privacy, which cannot be compared to DP. This is indeed not the case, thanks to the known implications between DP and f-DP, but I found the current presentation somewhat confusing regarding this point, and I recommend the authors clarify it in the final version.

I found some of the choices made in the numerical evaluations presented in Figure 1 to be not so clear. First of all, the sample size was chosen to 1,000 while the reference paper used 10,000, and $\delta$ was chosen as $1/n$ while it is often expected to be of the order $o(1/n)$.

A more substantial issue with the comparison presented in Figure 1 has to do with the presented baseline. If I understand correctly, the baseline results were chosen to be those of the closed-form presented in [20], but that work contains tighter results presented in Theorem III2, which can be numerically evaluated, and should be presented as the more relevant comparison.

Minor comments:

* In line 104, the inequality if a functional inequality, but this notion was not presented before, so the notation is somewhat confusing.

=======

**Edit after rebuttal discussion:**

The authors response satisfied my remaining concerns.

**Questions:**

i will appreciate some clarification regarding the choices made in the evaluation presented in figure 1, and the chosen baseline.

---

> ### Author Rebuttal · Authors · 2023-08-07
>
> We greatly appreciate your extremely positive rating and valuable comments. Thanks to your feedback, we revisited the numerical results of Feldman et al. (2023) and compared them with our own. The comparison showed that our methods outperformed their numerical upper bound. Additionally, we conducted a comparison with their numerical lower bound, which, to the best of our understanding based on their code, is obtained using binary search for privacy auditing. The results demonstrated that our bound is near-optimal. The reference number here is aligned with the supplementary materials. Below, we provide detailed responses.
>
> Responses to the weaknesses:
>
>
> 1. The choices in Figure 1. We initially used a sample size of $n=1000$ and set $\delta = 1/n$ as a simple example to demonstrate the sharpness of our $f$-DP bound. In response to the reviewer's suggestion, we have now updated Figure 1 and Table 1 with $n=10000$ and $\delta = o(1/n)$ which is in line with exisiting literature [23]. Please refer to the attached PDF for both the updated figure (Figure 5) and the new table (Table 3&4). Here, we display a simplified table with $\epsilon_0 = \log\left(\frac{n}{8\log (2/\delta)} - 1 \right) = 4.444$, which is the critical value required by Theorem 3.2 of Feldman et al. (2023) with $\delta = 10^{-6}$. It is worth noting that this assumption, caused by Hoeffding's inequality, is removed in our $f$-DP analysis, as we utilize Lemma 4.1 instead of Hoeffding's inequality. As seen in the updated table, our bound still significantly outperforms previous results.
>
> Table 3 in the attached pdf:
>
>
> $\epsilon$:   0.5             |      0.6     |    0.7    |    0.8     | 0.9       ｜ 1.0            ｜1.1
>
>  $\delta$ in [23]:  0.9494  |  0.3764 | 0.1038 | 0.0181 ｜0.0018 | $8 * 10^{-5}$| $2 * 10^{-6}$
>
>  $\delta_{f-DP}$:  $3 * 10^{-6} |  10^{-7}  |   4* 10^{-9} |  9 * 10^{-11} | 2 * 10^{-12}| 2 * 10^{-14}| 3* 10^{-16}$
>
>
> 2. Thank your for pointing out the minor errors in line 104. The functional inequality means the inequality holds pointwisely, which is pointed out in the revised version.
>
>
> Responses to the weaknesses:
>
>
> 3.  Choosing the numerical upper bound in [23] as the baseline. In the submitted version of our paper, we solely compared our results with the theoretical upper bound presented in [23]. However, following the reviewer's suggestion, we reevaluated the numerical results in [23] and conducted a more comprehensive comparison with their work. For this comparison, we selected $n=10000$, and $\epsilon_0$ was set to $4.444$, aligned with Item 1. The results of this comparison are listed in Table 4 in the attached PDF and we listed the simplified one below. It is important to note that for very small values of $\delta$, the numerical results in [23] remain invalid due to their assumptions on $\epsilon_0$ in their Theorem 3.2.
>
>      Additionally, we also compared our results with the numerical lower bound generated by inference attacks in [23]. This comparison further substantiates that our $f$-DP analysis is indeed near-optimal. From a theoretical standpoint, this near-optimality can be interpreted by considering the basis of our proofs. The proof of Theorem 3.1 relies on a post-processing procedure in [23], the joint concavity (Lemma 4.1 and Proposition 4.3), and the advanced joint concavity (Proposition 4.8). The post-processing procedure is sharp for specific mechanisms, such as the randomized response mechanism, as demonstrated by Theorem 5.2 and Theorem 5.3 in [23]. Proposition 4.3 also exhibits sharpness, as highlighted in Section 4. Furthermore, based on the proof of Proposition 4.8 in Section B.1.2, the advanced joint concavity utilized in this example is nearly the tightest closed-form bound attainable.
>
> Table 4 in the attached pdf:
>
>
> $\delta$:  $5 \times 10^{-5}| 3 \times 10^{-6} | 10^{-7}| 4 \times 10^{-9}| 9 \times 10^{-11}| 2 \times 10^{-12}| 2 \times 10^{-14}$
>
> $\epsilon_{fdp}$ (ours):   $0.4 | 0.5 | 0.6 | 0.7 | 0.8 | 0.9 | 1.0$
>
>   Numerical $\epsilon$ upper bound in [23]:  $1.014  |  1.085 | \epsilon_0 | \epsilon_0 | \epsilon_0 | \epsilon_0 | \epsilon_0$
>
>    Numerical $\epsilon$ lower bound:  $0.369 | 0.470 | 0.575 | 0.664 | 0.758 | 0.845| 0.944$

---

> > ### Comment · Reviewer_wgR7 · 2023-08-15
> >
> > I thank the authors for the clarifications and updates which satisfied my remaining concerns, and apologize for the late response.

---

> > > ### Author Response · Authors · 2023-08-15
> > >
> > > Thank you once more for your profoundly insightful feedback, particularly for prompting us to reevaluate the numerical findings presented by Feldman et al.

---

### Official Review · Reviewer_oyTp · 2023-07-06

**Soundness:** 3 good
**Presentation:** 3 good
**Contribution:** 3 good
**Rating:** 6
**Confidence:** 3

**Summary:**

Randomization is an essential too in deriving differentially private algorithms. However, sophisticated randomization techniques such as shuffling induce complicated distributions on outputs, making analysis of privacy loss difficult. The paper uses the framework of f-differential privacy to provide tighter analysis on the privacy guarantees of shuffled DP-SGD than existing methods do. Additionally, the authors show that randomized initialization can beneficially improve privacy guarantees for one step problems.

**Strengths:**

-	The closed-form bounds for shuffled DP-SGD seem particularly important, even if the bounds are complicated. Table 1 does a good job of convincing the reader that the obtained bounds are significantly better than existing ones.
-	Moreover, the authors present some useful results for studying tradeoff functions in the f-DP framework. These will likely be useful for future investigations.


**Weaknesses:**

-	Results for random initialization for one-step SGD do not seem particularly useful? In particular, this contribution seems like a bit of a toy problem.
-	Some demonstration of the improved bounds for Shuffle-SGD would be useful to the reader. In particular, it would be convincing if there was some simple learning task on which the accuracy of the model produced by shuffle-SGD (under the new analysis) significantly outperformed the model produced under the old analysis. This should happen for any experiment, as the privacy budget savings seems to be significant per table 1.


**Questions:**

NA

**Limitations:**

-	The authors fairly discuss the limitations of their work, in particular their intentions to study multi-step SGD with random initialization.

---

> ### Author Rebuttal · Authors · 2023-08-06
>
> We sincerely appreciate your positive comments and acknowledgment of the novelty of our results. In response to the identified weaknesses, we provide the following explanations:
>
> 1. Response to the first item. In Section 3.2, we applied Theorem 3.5 to analyze privacy amplification in noiseless 1-dimensional linear models, where the sensitivity of the gradient $\mu_I$ with a given initialization $I$ can be specified. It is worth noting that similar results are applicable to multidimensional linear models, where $y_i = a^T x_i$ and $x_ix_i^T = \mathbb{I}$ with $\mathbb{I}$ being the identity matrix. Noiseless linear models find relevance in various applications, as exemplified in the work by Berthier et al. (2020) titled "Tight Nonparametric Convergence Rates for Stochastic Gradient Descent under the Noiseless Linear Model."
>
>      For more general applications, such as general convex loss functions, deriving $\mu_I$ becomes more intricate and deserves further investigation, which we plan to address in our future study on the privacy of multi-step DP-SGD.
>
> Overall, the DP-GD part is not as promising as our results on shuffling models but it is still novel with applications to specific examples.
>
> 2. Response to the second item. We are grateful for your valuable suggestions regarding the experiments. As mentioned by Feldman et al. (2020), it is indeed possible to analyze the privacy of shuffled DP-SGD when the number of iterations is given. However, despite extensive literature review on shuffling models, we have not come across any experimental results involving running shuffled DP-SGD. We have compiled a list of related papers, including a survey paper, but due to the constraints of time for this rebuttal, we intend to present our experimental results for shuffled DP-SGD in a subsequent study.
>
>
>     To make our theory more convincing, we added one paragraph in Section 3.1 regarding the tightness of Theorem 3.1 as follows: The bound in Theorem 3.1 is near-optimal in theory, and it represents a significant enhancement compared to [23], as demonstrated in Table 1 and Figure 1. In fact, the proof of Theorem 3.1 is based on a post-processing procedure in [23], joint concavity (Proposition 4.3), and advanced joint concavity (Proposition 4.8). The post-processing procedure is sharp for specific mechanisms, such as the randomized response mechanism, as shown by Theorem 5.2 and Theorem 5.3 in [23]. Proposition 4.3 is also sharp, as demonstrated in Section 4. Based on the proof of Proposition 4.8 in Section B.1.2, the advanced joint concavity used in this example is almost the tightest closed-form bound that can be derived. Compared to [23], the main advantage of using $f$-DP (Proposition 4.3) is that we avoid the use of Hoeffding's inequality, which is adopted in [22, 23] and leads to loose bounds, to bound the mixture of binomial distributions. Moreover, Theorem 3.2 in [23] holds with an assumption on the local privacy budget $\epsilon_0$, which is removed by using $f$-DP in our paper.
>
>     Besides, we compared our upper bound with the numerical lower bound obtained by inference attacks which shows that our bound is near-optimal, as shown in Table 4 in the attached pdf.
>
>
> https://arxiv.org/pdf/2208.04591.pdf
>
> https://arxiv.org/pdf/2012.12803.pdf
>
> https://arxiv.org/pdf/2107.11839.pdf
>
> https://arxiv.org/pdf/2303.07160.pdf
>
> https://arxiv.org/abs/2105.05180
>
> http://proceedings.mlr.press/v139/ghazi21a/ghazi21a.pdf

---

### Official Review · Reviewer_s9n5 · 2023-07-06

**Soundness:** 2 fair
**Presentation:** 2 fair
**Contribution:** 2 fair
**Rating:** 3
**Confidence:** 4

**Summary:**

This paper studies an important problem in DP: the privacy quantification based on a mixture of randomness. Based on f-DP, the authors point out the joint concavity of the tradeoff function. Two potential examples are proposed, including the shuffling model and the privacy amplification from random initialization.

**Strengths:**

+ The problem studied is important to the privacy analyses of many applications. In particular, I really like the idea to consider the privacy amplification from the random initialization.

+ The introduction and the potential limitation are nicely written. Motivation is clear.

+ The improvement upon the privacy analysis on shuffling model seem interesting.

**Weaknesses:**

- Though I really like the idea to take the randomness of initialization into account and study its privacy implication, the results presented in Section 3.2 are not convincing. The claimed simulatable f-DP guarantees seem to be instance-based, or is with respect to a given adjacent datasets $D_0$ and $D_1$. Moreover, even for the particular instance-based guarantee, there is no analysis provided, such as high confidence bound, to allow us really apply this bound to produce rigorous privacy guarantee. Moreover, (please correct me if I get it wrong), based on my understanding, applying the joint concavity, Theorem 3.5 is essentially determined by the average case of the "local" sensitivity given the random initialization. So, I guess current studies cannot be generalized to the standard worst-case DP guarantee.

- The notations are confusing. For example, I did not find the formal definition of $(\theta(D)|I,I)$ in Section 3.2. My best guessing is because there are two random sources? But why we must assume the noise and random initialization are in the same distribution?

-  Section 3.1 is lack of theoretical analysis on how tighter the f-DP bound given in Theorem 3.1 is, as compared to previous works. Though the empirical improvement on the $\delta$ numbers seem significant, given that the dependence of $\epsilon$ on $\delta$ is in logarithm, it is not very clear whether such improvement is general. Another missing issue is the explanation on the computational accuracy $10^{-6}$ in Table 1. Where is this restriction coming from? Moreover, I think one major motivation to introduce f-DP in [14] is for tighter composition. The authors may also want to take it into consideration.

**Questions:**

1. How to really apply the privacy amplification results in 3.2 under regular input-independent f-DP scenario?

2. What the fundamental challenge to generalize the results to the subsampling case? I think the subsampled f-DP is already solved in "Deep Learning with Gaussian Differential Privacy" (https://arxiv.org/abs/1911.11607).

3. For the theoretical analysis, can you characterize the improvement compared to previous works?

4. It would be more helpful if the authors can elaborate the background of shuffling model, at least why the output distributions from two adjacent datasets are in the form described at the beginning of Section 4.


**Limitations:**

## Update after reading the authors responses:
I summarize my concerns and suggestions for the authors to improve this paper:
1. As I said, I always like the idea of privacy amplification from random initialization and this is an important open question widely-recognized in DP community. But the authors's results on this part, from my opinion, is trivial given that it can only study closed-form iterate distribution for the first round and thus negligible amplification in DP-SGD, and the authors agreed to put it as a minor contribution in the revision.

2. I think some claims in the paper are overblown and the limitations of the proposed methods are not properly discussed. At least to me, the joint convexity thing in the tradeoff function is not that surprising given that Poisson subsampling has already been studied in Gaussian DP. The authors also agreed that their improvement over the subsampling case is limited either for i.i.d. sampling or with a fixed batch size.

3. For the main contribution to the shuffling model, I never say that there is no novelty in this paper. I agree that the paper presents tighter bounds on some special cases, for example with additional assumptions on the ordering tradeoff function or special mixture weights. But for the shuffling model, the advantage is only measured empirically without a clear asymptotic analysis. As I mentioned, a small constant improvement on the $\epsilon$ can lead to exponential improvement on $\delta$. The authors' response on this part is somewhat double talk and constantly say they simulate it and it is better than prior works. But my question is always how better it is and the authors did not  directly give an answer to show analytical bound of improvement. Anyway, I suggest the authors clearly discussing about both the advantages and limitations.

---

> ### Author Rebuttal · Authors · 2023-08-06
>
> Thank you for your valuable feedback on our paper and insightful comments. We have carefully considered the raised questions and weaknesses and offer the following responses and modifications. The reference number here is aligned with the supplementary materials.
>
> Responses to weaknesses:
> 1. The first item. Section 3.2 is convincing but it has some restrictions, and in the revised version, we have added one paragraph discussing extension of Theorem 3.5 to the worst case and its restrictions:  In Theorem 3.5, given the initialization $I$, the sensitivity of the gradient $\mu_I = \mu_I(D_0, D_1)$ depends on choices of two neighboring datasets $D_0$ and $D_1$. To extend this to the worst-case, we consider the noiseless linear models with least squares loss (Example 3.3 and 3.4), which is also used in practice (https://arxiv.org/abs/2006.08212).
> In fact, $\mu_I (D_0,D_1)\equiv a - I$ when $D_1$ is constructed by deleting $\bf{arbitrary}$ element in $D_0$.
>
>     For more complicated models, such as those with general convex loss functions, the analysis of $\mu_I$ is involved and will be addressed in our future study of DP-SGD. As discussed in the end of this section, even for this simplest linear model, we observe privacy amplification while R'enyi DP fails to account for the privacy amplification.
>
>
>      Overall, the DP-GD part is not as promising as our results on shuffling models but it is still novel and is convincing with specific applications.
>
> 2. The second item. We acknowledged the potential misinterpretation of notations and have added an explanation for this notation in Sections 3 and similar notations in Section 4. Specifically, $(\theta(D)|I, I)$ denotes the observation of both $\theta(D)|I=i\sim\mathcal{N}(\mu_i,1)$ and the initialization (index) $I$. This is a relaxation of observing the mixture distribution $\theta(D)$ (a mixture of $\mathcal{N}(\mu_i,1)$ as stated in Line 167-168) without knowing the information of the index $I$. As observing the index exposes more information, distinguishing $\theta(D_0)$ and $\theta(D_1)$ is harder than distinguishing $(\theta(D_0)|I, I)$ and $(\theta(D_1)|I, I)$, which motivates Lemma 4.1 and Theorem 3.5. Releasing the index is essential to make the mixture distributions distinguishable (e.g., Example F.4 is not distinguishable without releasing indices). Moreover, we do not assume that $\theta(D)|I$ has the same distribution as $I$, although the initialization is usually Gaussian in practice. The expectation in Theorem 3.5 is taken w.r.t. $I$ and $I$ can be any distribution.
>
> 3. The first question of the third item. The tightness of Theorem 3.1 was discussed in Section 4 in the submitted version and we have added one more paragraph in Section 3.1 discussing the tightness in theory in the revised version.
> The bound is near-optimal in theory, and it represents a significant enhancement compared to [23].The proof of Theorem 3.1 is based on a post-processing procedure in [23], the joint concavity (Proposition 4.3), and the advanced joint concavity (Proposition 4.8). The post-processing procedure is sharp for specific mechanisms, such as the randomized response mechanism, as shown by Theorem 5.2 and Theorem 5.3 in [23]. Proposition 4.3 is also sharp, as demonstrated in Section 4. Based on the proof of Proposition 4.8 in Section B.1.2 before line 542, the advanced joint concavity used in this example is almost the tightest closed-form bound that can be derived. Compared to [23], the main advantage of using $f$-DP (Proposition 4.3) is that we avoid the use of Hoeffding's inequality and Chernoff bound to bound the mixture of binomial distributions, which is adopted in [22, 23] and leads to loose bounds.
> Moreover, Theorem 3.2 in [23] holds with an assumption on the local privacy budget $\epsilon_0$ which is removed by using $f$-DP in our paper.
> Additionally, based on Table 3 and 4 in the pdf attached, our bound is near-optimal as it is close to the numerical lower bound computed by privacy auditing.
>
> 4.  The second question in the third item. The accuracy $10^{-6}$ was due to calculating the dual function in Matlab. We have refined our code, and now the accuracy can be $10^{-20}$ as shown in the attached pdf.
>
> 5. The third question in the third item. The composition property is a fundamental aspect of differential privacy (DP) and composition + shuffling will also be investigated in our future study. However, it differs from our current privacy analysis. Specifically, computing the privacy of DPSGD through composition requires releasing the parameters (models) output by every iteration while our primary focus lies in analyzing the privacy of the last-iterate parameter without disclosing intermediate ones (cf., [2,43] for the analysis using RDP). Although our present paper only examines the one-step iteration, we are actively extending it to DP-SGD in the future. This extension is highly non-trivial. The role of joint concavity (Lemma 4.1) is crucial, building on insights from the analysis in [43] for RDP.
>
> Responses to the questions:
> Question 1 & 3 are answered in Item 1 & 3 above.
>
> 6. Question 2: Besides the simplest sampled mechanism in [9,14], there are other sampling mechanisms such as Poisson sampling or sampling with replacement. For sub-sample mechanisms, one may refer to [4] for privacy analysis, where the joint convexity and advanced joint convexity of the hockey-stick divergence play important roles. Our Lemma 4.1 and Lemma 4.7 are extensions from the hockey-stick divergence to trade-off functions.
>
> 7. Question 4: We have taken the reviewer's suggestion and added a detailed citation (Theorem 3.2 in [23]) and provided clear explanations at the beginning of Section 4 to better elucidate the post-processing procedure. As we mentioned earlier, this post-processing technique from [23] is highly effective for specific mechanisms, making our $f$-DP bound near-optimal.

---

> > ### Comment · Reviewer_s9n5 · 2023-08-13
> >
> > I would like to thank the reviewers' response. I carefully read the reply and also authors' responses to the other reviewers. But still, my two main concerns about the paper's novelty remains.
> >
> > First,  I do not think results in section 3.2 are meaningful, or at least the author should not claim this as a main result of this paper. The results are far away from solving problem of privacy amplification from random initialization. At most, the authors' methods only address the DP analysis for running DP-SGD on very simple quadratic linear regression (gradient has a closed form) and for one iteration. The results are trivial since even without using the tradeoff function, the iterate distribution already enjoys a closed form by just combing the Gaussian noise and initialization. That is why I state once the loss function is a bit more complicated, the methods proposed can only handle very weak instance-based DP then.
> >
> > Second, I guess I may not express my concerns on the subsampling issues very clearly. My point is that subsampling is a classic distribution mixture model and many existing works studying f-DP for subsampling have also essentially studied the f-DP on mixture distributions. Compared to the random initialization model, subsampling is definitely a more well-studied and better-understood model. At least, I do not see any straightforward application of the paper's results on mixture distribution to produce sharpened results on the subsampling amplification.
> >
> > As a summary, as the title of the paper is a "unified analysis" of f-DP on mixture models, I do not think the methods presented here are generic or novel enough to handle all the applications. So, I decide to keep my score.

---

> > > ### Author Response · Authors · 2023-08-14
> > > **Response to the reviewer, I**
> > >
> > > We thank the reviewer for clarifying questions in the official comments that is not specific in the initial reviews.
> > > The weakness and novelty of our paper were stressed in the rebuttal clearly. I would like to strengthen our novelty, contributions, and limitations again: we provide a unified $f$-DP analysis framework for mixture mechanisms, that applies to shuffling models and DP-GD with random initialization (with some strong assumptions on the models to extend it to the worst case). This framework can also be extended to the shuffled and sub-sampled multi-step DP-SGD which will be investigated in our future study.
> > >
> > >
> > > 1. In response to the reviewer's inquiries concerning the application of the least squares loss in the context of privacy amplification with random initialization, we offer the following explanation:
> > >
> > >
> > >      First, we extend our findings to a worst-case study and utilize linear logistic regression as an illustrative example. In general, achieving the worst case DP guarantee is to replace $\mu_I = \mu_I(D_0,D_1)$ in Theorem 3.5 by the worst case sensitivity $\mu_I^*$ with a given initialization $I$, where $\mu_I^*$ is such that $|\mu_I^*| = \max_{D_0, D_1} |\mu_I(D_0,D_1)|$.  We now use linear logistic regression as an example to specify the worst case sensitivity $\mu_I^*$. Let $D_0$ be an arbitrary dataset and let $D_1$ be generated by deleting $z = (x,y)\in D_0.$ Then, we have $\mu_I(D_0,D_1) = \frac{\eta e^{-yI\cdot x}}{1 + e^{-yI \cdot x}}$ and worst case sensitivity is $\mu_I^*$ is such that $|\mu_I^*| = \sup_{x,y} \left|\frac{\eta e^{-I\cdot yx}}{1 + e^{-I \cdot yx}} \right|$ as the gradient of the logistic loss is the softmax function $\frac{e^{-I\cdot yx}}{1 + e^{-I \cdot yx}}$.
> > >
> > >     Let's consider the noiseless linear model discussed in https://arxiv.org/abs/2006.08212, where we assume $y = ax$ and $x^2 = 1$ for some $a\neq 0$. In this context, the worst-case sensitivity $\mu_I^*$ is given by $\mu_I^* = \frac{\eta e^{-aI}}{1 + e^{-aI}}$. The numerical computation of the worst-case trade-off function can be performed by substituting $\mu_I$ in Theorem 3.5 with $\mu_I^*$. By comparing the resulting trade-off function with respect to $\mu_I^*$ to the that of $1$-GDP (the sensitivity of the softmax function is $1$), we observe the privacy amplification achieved through random initialization. In general, the impact of initialization in worst-case differential privacy (DP) can be examined under the condition that $|xy| \leq M$  for some $M > 0$. Note the the key factor is the monotonicity of the softmax function. Similar results may hold when the gradient is monotonic, which holds for convex generalized linear models (cf., https://arxiv.org/pdf/2006.06783.pdf.)
> > >
> > >
> > >    In response to your inquiries regarding the utilization of the least squares loss, we offer the following specific explanation:
> > >
> > >     In general, the initialization $I$ is not restricted to any specific distribution, thus $\mu_I$ may not inherently follow a Gaussian distribution. Take Example 3.4, which represents a worst-case scenario achieved by removing an arbitrary element from a dataset, while applying gradient clipping with a constant $c$. Here, the initialization $I$ is truncated as a result of its appearance in the gradient, leading to a distribution that is not characterized by a Gaussian + Gaussian combination, making its verification challenging. Our Figure 2 visually demonstrates the observed privacy amplification using our Theorem 3.5 when compared to $c$-GDP without considering the random initialization.
> > >
> > >     Example 3.3, in the absence of gradient clipping, indeed corresponds to a Gaussian + Gaussian distribution. The trade-off function involves testing two Gaussian distributions characterized by distinct means and variances. It's noteworthy that the closed-form expressions for type I and type II errors exist. However, representing the trade-off function in a closed form is only feasible when both distributions share the same mean. We greatly appreciate your insightful observations. Moroever, we intend to relocate this example to the appendix in the revised version of our work.
> > >
> > >     Overall, strong assumptions on the models are required for us to extend it to the worst case but it is still new and not demonstrated in existing results. As a result, we finally decided to keep this section with more discussions on its restrictions. We have highlighted this restriction in both the abstract and the introduction. Technically, this analysis of initialization is non-trivial and is a result of our $f$-DP framework (Lemma 4.1). Moreover, we hope our analysis serves as an important first step for people to notice the effect of initialization on DP analysis. Additionally, this is not claimed as our main contribution in the revised version and our main contribution is refining the bound for the shuffling models.

---

> > > > ### Author Response · Authors · 2023-08-14
> > > > **Response to the reviewer, II**
> > > >
> > > > 2. I interpret your second question as, since sub-sampling mechanisms has already been investigated using $f$-DP in "Deep learning with GDP" as you referenced in the rebuttal, you don't approve of the word "unified" in the title as our conclusion may not be more general than theirs. We would like to make the following response.
> > > >
> > > >
> > > >     In the paper "Deep Learning with Gaussian Differential Privacy", the sub-sampled mechanism investigated is the simplest Poisson sampling as their analysis is based on the central limit theorem. Our Lemma 4.7 is a more general result compared to their analysis and leads to the same $f$-DP sub-sampling bound as theirs in this simple case.  However, as this case is too simple, our Lemma 4.1 and the corresponding results are not studied and used in their paper.
> > > >
> > > >     Since our ambition is to count the privacy of the last-iterate parameters, we aims to analyze "fixed batch-size sub-sampling with replacement for multiple iterations" that combines privacy amplification by iteration + sub-sampling + shuffling, and, to the best of our knowledge, has $\textbf{NOT}$ been explored using $f$-DP in any existing literature and is much more complicated than the privacy analysis of the simplest case in the paper "Deep Learning with Gaussian Differential Privacy".
> > > >
> > > >     Concerning the $f$-DP analysis of more complicated sampling with replacement, our $f$-DP framework (both Lemma 4.1 and Lemma 4.7) should be used to extend the systematically study of sub-sampling with $(\epsilon,\delta)$-DP in Privacy Amplification by Subsampling: Tight Analyses via Couplings and Divergences by Balle et al.  This extension is straightforward using our Lemma 4.1 and Lemma 4.7 as our Lemma 4.7 is a generalization of their Theorem 2.
> > > >
> > > > We use "unified" in the title to describe our framework (Lemma 4.1 and Lemma 4.7) as it implies the joint convexity of $F$-divergence (such as the exponentiation of the R\'enyi divergence) for any $F$ and this joint convexity can be found in almost all existing literature regarding the privacy analysis of  mixture mechanisms including the privacy amplification by iteration, shuffling, and sub-sampling.
> > > > On the other hand, the joint convexity of $F$-divergence doesn't not imply our Lemma 4.1 as pointed out by our paper.
> > > > Thus, our Lemma 4.1 is indeed more general than any divergence-based DP.
> > > >
> > > > One may refer to Lemma 2.6 in https://arxiv.org/pdf/2205.13710.pdf  and Lemma 4.1 in https://arxiv.org/abs/2203.05363 for the applications of the quasi-convexity of R\'enyi divergence  and the joint convexity of the exponentiation of the R\'enyi divergence to privacy amplification by iterations bounds. As those kind of convexity notions of $F$-divergences are special cases of our Lemma 4.1 and Lemma 4.7, we believe our results can be used to extend existing analysis of mixture mechanisms based on RDP or $(\epsilon,\delta)$-DP to $f$-DP.
> > > > Thus, we keep "unified" in the title.

---

> > > ### Author Response · Authors · 2023-08-14
> > >
> > > Our most significant contribution is in the realm of shuffling models, where we establish a substantial enhancement compared to the state-of-the-art results. Moreover, this bound is nearly optimal when compared with the empirical lower bound. This contribution is both novel and significant. The shuffling procedure is widely adopted in training deep learning models, as can be seen from both TensorFlow and PyTorch (Opacus). We would like to emphasize once more this main contribution, as the reviewer didn't mention this part as our main contribution in the reviews.
> > >
> > > Overall, a rating of 3 might arguably be worth a second thought as our results do present near-optimal analysis for the widely used shuffling models.

---

> > > > ### Comment · Reviewer_s9n5 · 2023-08-14
> > > >
> > > > Thanks for your prompt reply.
> > > >
> > > > First, I think no prior works on privacy amplification from random initialization like the one-round analysis proposed in this paper is mainly because even for the simple strongly-convex case mentioned by the authors, such improvement is negligible. The really challenging thing is to show the amplification from the trajectory.
> > > >
> > > > Second, I am not against studying the subsampling with/out replacement and I did not say that this work must apply to the Poisson case. But the bottom line is there is absence of discussions to the sampling with fixed batchsize or how the results can improve, such as, Balle et al's prior works. My point is given that the authors claim this is an unified framework and in general f-DP will produce stronger composition to Renyi DP, it would not be hard to show very clear improvement over the subsampling amplification. But as I said, I did not see the results in Section 4 can apply to those applications, given the very special weight selection, such as that in Section 4.2.
> > > >
> > > > Third, I mentioned my concerns on the results for shuffling in the initial comments. There is no asymptotic or clear comparison with existing works, such as "Hiding Among the Clones: A Simple and Nearly Optimal Analysis of Privacy Amplification by Shuffling". I am afraid that simply presenting empirical results are not that convincing, especially only showing some cases of $\epsilon$ and $\delta$. The $\epsilon$ is in around the same order of  the log of $delta$. Thus, I express my concern that the improvement is not really clear to me.

---

> > > > > ### Author Response · Authors · 2023-08-14
> > > > >
> > > > > Thank you for your prompt new comments.
> > > > >
> > > > > 1. The reviewer's first new comment is different from the initial reviews, where the reviewer expressed interest in privacy amplification by random initialization. As we highlighted in our revised version, we assert this as our secondary contribution. We believe that our clarification regarding the extension to worst-case DP has been adequately conveyed in our previous responses.
> > > > >
> > > > >     Moreover, we also acknowledge that analyzing the trajectory is more important in the privacy analysis of DP-SGD, which is known as privacy amplification by iteration as we discussed. As the analysis of trajectory is interesting but is much more complicated than analyzing the initialization, we believe it deserves a separate dedicated study in the future.
> > > > >
> > > > > 2. Your perspective that $f$-DP has tight composition bounds holds in the $\textbf{asymptotic}$ sense since "Deep learning with Gaussian DP" is based on the central limit theorem (CLT) for an $n$-fold composite algorithm as $n\rightarrow +\infty$. Without sub-sampling, the convergence of CLT can be very fast based on some of our recent observations, which can also be seen from "A Central Limit Theorem for Differentially Private Query Answering" by Dong et al.
> > > > > However, in the existence of mixture such as sub-sampling, the trade-off function is far from Gaussian and the convergence of the CLT requires a huge $n$  (see some recent work "Analytical Composition of Differential Privacy via the Edgeworth Accountant" by Wang el al. using Edgeworth accountant).  In the $\textbf{more pratical non-asypmtotic}$ case with small $n$, CLT may under-estimate the privacy budget of subsampling (https://proceedings.mlr.press/v108/koskela20b.html).
> > > > > When $n$ is small, the $f$-DP guarantee for the composite mechanism is calculated by the tensor product of $n$ trade-off functions as shown by https://arxiv.org/pdf/1905.02383.pdf [9]. Except for GDP (a special case of $f$-DP), calculating the tensor product between two general trade-off functions is not easy without using the central limit theorem.
> > > > >
> > > > >     It is worth noting that the composition of DP mechanisms is different from the mixture of mechanisms. The mixture occurs when processing a dataset $\textbf{once}$ by multiple random mechanisms, which makes the dataset $\textbf{more private}$, while the composition property is to measure how privacy guarantees $\textbf{degrade}$  when the dataset is processed $\textbf{multiple times}$.
> > > > >
> > > > >         Thus, the reviewer's perspective that since GDP implies tight composition bound asymptotically,  $f$-DP should outperform $(\epsilon,\delta)$-DP in the analysis of mixture mechanisms easily is not accurate.
> > > > >
> > > > >     Go back to our framework where we aim to study the trade-off function for mixture mechanisms in the non-asymptotic sense. The sub-sampling results in Balle et al. have been proved to be tight in terms of $(\epsilon,\delta)$-DP. If we extend their analysis to $f$-DP using Lemma 4.7 and convert $f$-DP to $(\epsilon,\delta)$-DP, then we obtain the same $(\epsilon,\delta)$-DP  bound as Balle et al.  In response to the reviewer's questions about the weight selections presented in Lemma 4.7, we acknowledged that deriving the tightest weights may not lead to a closed-form and our choice of the weights for analyzing sub-sampling mechanisms is in line with Balle et al. which is near-optimal and is almost the tightest bound we can obtain.
> > > > >
> > > > >      However, $f$-DP is a completion of $(\epsilon,\delta)$-DP and is more informative than $(\epsilon,\delta)$-DP as shown by the original GDP paper [9]. If we compare our bound with Balle et al. in terms of $f$-DP, our bound is more informative as the trade-off function for $(\epsilon,\delta)$-DP  is linear while our $f$-DP bound is its completion and is non-linear. As can be seen from  [9], divergence-based DP (including both R\'enyi DP and $(\epsilon,\delta)$-DP) is not as informative as $f$-DP. Conclusions in $f$-DP (including our Lemma 4.1 and Lemma 4.7) imply the corresponding results in divergence-based DP while $F$-divergence can't be converted back to $f$-DP, as discussed in Section B.2 in [9].
> > > > >
> > > > >
> > > > >
> > > > >       In the analysis of simple basic sub-sampling, Lemma 4.7 is enough to achieve the privacy analysis. However, for complicated mixture mechanisms such as shuffling, Lemma 4.7 may fail as the weights can not be calculated.  To address this challenge, we introduce Lemma 4.1, which can be used in analyzing all mixture models. Lemma 4.1 is novel and is different from existing analysis in Balle et al., "Gaussian differential privacy" (https://arxiv.org/abs/1905.02383), and "Deep learning with Gaussian differential privacy" (https://arxiv.org/abs/1911.11607) you referenced. "Unified" means our framework can  analyze any mixture mechanisms non-asymptotically and is better than (or at least as good as) exisiting privacy analysis for extensive mixture mechanisms.

---

> > > > > ### Author Response · Authors · 2023-08-14
> > > > >
> > > > >
> > > > > 3. Regarding shuffling models, we compared with the state-of-the-art results https://arxiv.org/abs/2208.04591 [23] which are the most-up-to-date by Feldman el al. instead of the old one (https://arxiv.org/pdf/2012.12803.pdf ) [22] you mentioned. In the 1-page pdf attached, we compare with them in terms of both $\epsilon$ and $\delta$. The new choices of $n=10000$ and $\delta = o(n)$ is adopted in [22] as suggested by reviewer wgR7. In Figure  6 in the attached pdf, I compared the $\delta$ parameter with [23] for $\epsilon_0$ varying in an interval. We have illustrated our results using a new figure where we take log of the $\delta$ and show that our bound is much tighter. As we are not allowed to upload figures now, we will add it to the revised version. It can be seen clearly from our Table 3 that even taking log of the $\delta$ parameter, our bound is obviously much tighter.
> > > > >
> > > > >     We also explained the tightness of our results in general theory. While our Proposition 4.3 is tight compared to Lemma A.4 in SOTA, the post-processing and the use of Lemma 4.7 are in line with the $(\epsilon,\delta)$-case in [23]. Thus, our bound is always better than theirs.
> > > > >
> > > > > We appreciate your detailed comments that make our paper better, and we hope that all your concerns are clarified. Please let us know if you have any other concerns.

---

> ### Comment · Reviewer_wgR7 · 2023-08-15
>
> If I may ask the reviewer for clarifications myself.
>
> As I see it, the main contribution of the paper in Theorem 3.1 and the theoretical tools that were used to prove it (presented mainly in section 4). I agree with the reviewer that the current presentation paints a slightly different picture, and I recommend the authors align their final presentation with the assessment provided by the reviewers.
>
> That being said, I am still not sure I understand the reviewers opinion on this part's contribution. Is it just not important enough in their opinion (in which case, I don't have much to add, and it is probably up to the AC to decide), or do they also find a significant limitation in this result (such as not being convinced by the tightness argument). If it is the latter, I would appreciate a clarification of the gap, as they see it.

---

> > ### Author Response · Authors · 2023-08-16
> > **Response to the reviewers update.**
> >
> > The reviewer revised their reviews instead of adding official comments, almost causing us to overlook them. All questions in the reviewer's update have already been clearly addressed in our rebuttal, which seems to be overlooked by the reviewer. Here, we would like to address them again in response to the reviewer's update.
> >
> >
> > 1. Response to Item 1 in Limitations: we claim it as our secondary contribution doesn't mean it is trivial. As required by the reviewer, we extended our analysis to linear models with strongly convex loss in our rebuttal, which is non-trivial and can be found in, for example, https://arxiv.org/abs/1803.02596, https://arxiv.org/abs/2007.05157.
> >
> > 2. Response to Item 2 in Limitations: our framework is designed to deal with $\textbf{any}$ mixture models. Our Lemma 4.7 is motivated by the subsampling analysis in Balle et al. and is a generalization of their Theorem 2 to $f$-DP. $\textbf{For complicated mixture models such as shuffling,}$ $\textbf{Lemma 4.7 and their Theorem 2 may not work}$ due to the difficulty to clarify the weights. To deal with this challange,  we propose novel Lemma 4.1 that can handle $\textbf{any}$ mixture distributions and lead to $\textbf{sharp}$ analysis (Proposition 4.3) in the applications of the widely used shuffling models.
> >
> >
> > 3. Response to Item 3 in Limitations: in our rebuttal we already show that our bound is near-optimal $\textbf{ in theory}$ and is $\textbf{ always}$ better than SOTA (https://arxiv.org/pdf/2208.04591.pdf) instead of in some special cases as demonstrated by the reviewer.
> > Moreover, we $\textbf{removed}$ the assumption required by Theorem 3.2 in SOTA while the reviewer $\textbf{misunderstood}$ it as we add stronger assumptions.
> > Here we emphasize the tightness again in terms of the proof details that is in line with our rebuttal.
> >
> >     The proof of Theorem 3.1 is based on a post-processing procedure in SOTA, joint concavity (Proposition 4.3), and advanced joint concavity (Proposition 4.8). The post-processing procedure is sharp for specific mechanisms, such as the randomized response mechanism, as shown by Theorem 5.2 and Theorem 5.3 in SOTA. Proposition 4.3 is also sharp, as demonstrated in our Section 4. Based on the proof of Proposition 4.8 in Section B.1.2 (the content before line 542 is the proof of Proposition 4.8 and is clarified in the revised version), the advanced joint concavity used in this example is almost the tightest closed-form bound that can be derived. Compared to SOTA, the main advantage of using  $f$-DP (Proposition 4.3) is that we avoid the use of Hoeffding's inequality and the Chernoff bound, which is adopted in SOTA and leads to loose bounds, to bound the mixture of binomial distributions. Moreover, Theorem 3.2 in SOTA holds with an assumptions on the local privacy budget $\epsilon_0$ , $\textbf{which is removed by using Proposition 4.3 in our paper.}$
> >
> >     Since our novel Proposition 4.3 is sharp and the other details of the proof are in line with SOTA, our bound is always better than the SOTA. Both our bounds and that in SOTA are non-asymptotic bounds and the non-asymptotic analysis is more practical than asymptotic analysis. We already show that our bound is $\textbf{always better than SOTA in terms of tightness and assumptions}$. Comparing with them in the asymptotic sense is unnecessary.

---

> > > ### Comment · Reviewer_s9n5 · 2023-08-16
> > >
> > > Thank you very much for your response. I respect your perspective on the selection of results to be incorporated into the paper. However, it appears that several of our discussions might not be entirely aligned or on the same page, and I believe it's best to entrust the final decision to the AC.
> > >
> > > I want to emphasize that I acknowledge the authors' valuable efforts to enhancing the state-of-the-art (SOTA) methodologies. My previous remarks were solely centered around the absence of a comprehensive analytical exposition detailing these enhancements. Drawing from my own experience, I've observed significant disparities in the trade-off function across various parameter configurations, particularly in the realms of high and low privacy regimes. Consequently, I'm asserting that the empirical simulations provided do not offer me a clear grasp of the situation at hand. As alluded to in the authors' response, if they believe that the superiority of the proposed bound over the SOTA is self-evident in all scenarios, it would greatly enhance clarity to augment this with a thorough analysis. Such an analysis would serve to elucidate the advancements of the proposed bound to the reader, providing a clarity that surpasses mere simulations. Such analytical  approach holds particular value, given the infeasibility of enumerating all potential cases.
> > >
> > > Furthermore, I'd like to express my reservations about the omission of discussions pertaining to the applicability of the proposed techniques to other contexts. While the authors have mentioned various motivations in our discussions, without these being explicitly documented in the paper, I find myself unable to corroborate these assertions. Regardless, please accept my apologies if my comments inadvertently created any ambiguity or conveyed a sense that I dismissed the diligent efforts expended in deriving tighter bounds for the shuffling model or the more generalized applications. I am content to leave the ultimate judgment in the hands of the AC.

---

> > > > ### Author Response · Authors · 2023-08-17
> > > >
> > > > Thanks for the reviewer's valuable new comments.
> > > > While we respect the reviewer's view on the broader impact of our joint concavity framework, we maintain our stance on the importance of our $f$-DP framework concerning joint concavity (Lemma 4.1). This is highlighted by Proposition 4.5 in our paper, where Lemma 4.1 implies the joint convexity of $F$-divergence for any convex $F$, crucial for analyzing privacy amplification in DP-SGD (cf., Lemma 4.1 of https://arxiv.org/abs/2203.05363 and Lemma 2.6 of https://arxiv.org/abs/2205.13710). Furthermore, our Lemma 4.1 yields a tight bound in the shuffling model analysis, as demonstrated by Proposition 4.3. Its significance is underscored by its tightness and extensions. Nevertheless, we concur with the decision to rely on the AC's judgment.
> > > >
> > > > Here are our responses to the other comments provided by the reviewer:
> > > >
> > > > 1. $\textbf{Comments:}$ "Drawing from my own experience, I've observed significant disparities ..., particularly in the realms of high and low privacy regimes":
> > > >
> > > > $\textbf{Response:} $
> > > > We have conducted a diligent exploration of the existing literature and have not encountered similar observations to those made by the reviewer regarding the variations in trade-off functions under different privacy budgets.
> > > > Our understanding is that the reviewer may be considering the potential impact of different privacy budgets on the central limit theorem (CLT) with respect to composition (https://arxiv.org/abs/1911.11607) as the privacy budget might influence the convergence rate of the CLT as described in terms of the Edgeworth accountant (https://arxiv.org/abs/2206.04236).
> > > > However, as we clarified in our previous response, our analysis is exact and is non-asymptotic
> > > > which is regardless of the privacy budget. Even if we consider the non-asymptotic composition using tensor-product between trade-off functions,
> > > > there is no difference between low-privacy and high-privacy regimes (https://arxiv.org/abs/1905.02383).
> > > > Furthermore, it's worth noting that both $f$-DP analysis and other divergence-based DP analyses for mixture models, as indicated in the numerous sources we provided in our previous responses, consistently hold true for both high-privacy and low-privacy regimes.
> > > >
> > > > Additionally, we have noted that all our existing empirical results lie in the low-privacy regime with $\epsilon_0 > 1$. In order to address the reviewer's inquiry regarding comparisons between low-privacy and high-privacy regimes, we have conducted experimental details concerning the high-privacy scenario ($\epsilon_0 = 0.9$). Furthermore, we have heeded the reviewer's suggestion to replace $\delta$ with $1/\log(1/\delta)$.
> > > >
> > > > The outcomes of part of our experiments are outlined in the table below, and additional figures depicting results for varying privacy budgets will be provided in the revised version.
> > > >
> > > > | $\epsilon$  | 0.1  | 0.3  | 0.5  | 0.7  |
> > > > |---|---|---|---|---|
> > > > | $1/\log(1/\delta) (SOTA)$  | 0.2401  |  0.0165 | 0.0050  |  0.0020 |
> > > > |  $1/\log(1/\delta_{fDP}) (ours)$ |  0.0221 | 0.0037  | 0.001 | $ 10^{-10} $|
> > > >
> > > >
> > > > 2.
> > > > $\textbf{Comments:}$ "Consequently, I'm asserting that the empirical simulations ... at hand. As alluded to in the authors' response, if they believe ... a thorough analysis. Such analytical approach ... cases".
> > > >
> > > >  $\textbf{Response:}$
> > > > In addition to the empirical analysis, we have already provided a tightness analysis that we believe to be a "through analysis" in our rebuttal.
> > > > This analysis expounds on the $\textbf{tighter}$ nature of our bound under $\textbf{weaker}$ conditions compared to the SOTA (https://arxiv.org/pdf/2208.04591.pdf). We achieve this by delving into the $\textbf{proof details}$, particularly highlighting our Proposition 4.3. This proposition $\textbf{sharply}$
> > > > enhances Lemma A.4 in SOTA, while the rest of our analyses align with the SOTA.
> > > > We welcome any specific feedback from the reviewer that points out areas in our tightness argument that might be lacking in solidity.
> > > >
> > > > We are uncertain about the specific instances where the term "alluded" in the reviewer's comments is used, as our proof sketches and tightness analysis are clear. Identifying the specific instances where the term "alluded" is applied would be helpful, enabling us to provide additional details if necessary.
> > > >
> > > >
> > > > The term "analytical approach" might have also confused us. Our current interpretation of an "analytical approach" is that the reviewer suggested we involve the proof of an $\textbf{mathematical inequality}$ to compare  our Corollary 3.2 with the SOTA.
> > > > We note this challenge since our Corollary 3.2 is a near-optimal closed form and thus too complicated, as mentioned in Section 3, and is not as concise as the SOTA. A theoretical tighter DP bound for complicated mixture models is typically intricate (cf., Theorem 11 in https://arxiv.org/pdf/1807.01647.pdf). We'll highlight the challenge of comparing with SOTA analytically in our paper, despite our comprehensive tightness analysis.

---

### Official Review · Reviewer_jEfn · 2023-07-09

**Soundness:** 3 good
**Presentation:** 3 good
**Contribution:** 3 good
**Rating:** 5
**Confidence:** 2

**Summary:**

This paper studies improving the analysis of privacy bounds for two randomization processes (privacy amplifications): shuffling models (where each user record is privatized by some local randomizer like randomized response mechanism) and differentially-private gradient descent (DP-GD, where a Gaussian noise is added to the gradient update at each iteration). Previous bounds were given for the standard $(\epsilon, \delta)$-DP bound, while this paper analyzes through $f$-DP, a differential-privacy bound considering the trade-off between type I and type II errors (like the $f$-score), using some joint convexity arguments.

When translating the $f$-DP bounds back to bounds on $\epsilon$ and $\delta$, this paper gets much tighter bounds that works for more general ranges of $\epsilon$ and $\delta$.

**Strengths:**

The improvement in DP bounds are non-negligible and more general, and the presentation is convincing.

**Weaknesses:**

Certain critical aspects of the proofs might be better explained: the proof depends on some close-form expressions of piecewise linear functions (Theorem 3.1, Corollary 3.2, Proposition 4.3), which might have meaning beyond just those arithmetic manipulations or expressions as is currently shown in the paper. A one- or two-sentence description, or some simple examples, might help here.

**Questions:**

After Proposition 4.8, the paper writes “One may refer to the supplementary materials for details”, but this reader is unsure where are the supplementary materials.

**Limitations:**

The theoretical results in this paper does not have broader societal impacts.

---

> ### Author Rebuttal · Authors · 2023-08-05
>
> Thank you for your positive comments and valuable suggestions regarding our paper, particularly in acknowledging our main contributions. In response to the raised questions and weaknesses, we would like to provide the following explanations:
>
> Response to the weakness:
>
> In the revised version, we have explained the expressions of the trade-off functions derived in Theorem 3.1 in Section 3.1. Additionally, we have provided a comprehensive explanation of the trade-off function for testing two discrete distributions using the Neyman-Pearson lemma, employing the simple Bernoulli distribution as an example in Appendix A.
> The trade-off function for testing two discrete distributions is known to be piecewise linear, a well-established result based on the Neyman-Pearson lemma. In this context, the slope of each line segment is $-Prob_{X\sim Q}[p(X)/q(X) = t]/Prob_{X\sim P}[p(X)/q(X) = t],$ which is constant as $t$ only relies on the knots, in the Neyman-Pearson lemma.
> For example, for testing two Bernoulli distributions $Bern(p_0) v.s. Bern(p_1)$ with $p_1 > p_0$, the trade-off function is piecewise linear with 3 knots: $(0,1), (p_0, 1 - p_1)$, and $(1,0)$.
> Since the shuffling procedure introduces the mixture of binomial noise, which is a discrete distribution, the trade-off functions presented in Theorem 3.1, Corollary 3.2, and Proposition 4.3 are also piece-wise linear with numerous knots due to the mixture.
>
>
> Response to the Question:
>
>
> We acknowledge the confusion regarding the presentation of the proof of Proposition 4.8, which is contained implicitly in the proof of Theorem 3.1 in Section B.1.2 in the submitted version.
> Precisely, in Section B.1.2, the content before line 542 is the proof of Proposition 4.8.
> To address this, we have made the necessary modifications. Specifically, we have updated the title of Section B.1 to "Proofs of Proposition 4.3, Proposition 4.8, and Theorem 3.1." Additionally, we have divided Section B.1.2 into two separate sections, one focusing on the proof of Proposition 4.8 and the other on the proof of Theorem 3.1. This reorganization aims to improve clarity and facilitate better understanding for the readers.

---

> > ### Comment · Reviewer_jEfn · 2023-08-20
> >
> > Thank you for your rebuttal.
> >
> > I think other reviewers have more to add to the discussion, and I may just maintain my original ratings.

---

### Author Rebuttal · Authors · 2023-08-08

We greatly appreciate the reviews from all the reviewers, which have contributed to the thoroughness of our paper. We have attached a 1-page PDF here that provides detailed comparisons and extra numerical results.
Furthermore, we have noticed several frequently asked questions and would like to emphasize our responses as follows. Here the reference number is aligned with the supplementary materials, which is slightly different from the 9-page paper due to the appendix.

1. The tightness of Theorem 3.1 for shuffling models: Our Theorem 3.1 offers a near-optimal bound.

    Theoretically, we have added one paragraph right behind Theorem 3.1 to illustrate its near-optimality:

    The proof of Theorem 3.1 is based on a post-processing procedure in [23], joint concavity (Proposition 4.3), and advanced joint concavity (Proposition 4.8). The post-processing procedure is sharp for specific mechanisms, such as the randomized response mechanism, as shown by Theorem 5.2 and Theorem 5.3 in [23]. Proposition 4.3 is also sharp, as demonstrated in Section 4. Based on the proof of Proposition 4.8 in Section B.1.2 (the content before line 542 is the proof of Proposition 4.8 and is clarified in the revised version), the advanced joint concavity used in this example is almost the tightest closed-form bound that can be derived. Compared to [23], the main advantage of using $f$-DP (Proposition 4.3) is that we avoid the use of Hoeffding's inequality and the Chernoff bound, which is adopted in [22, 23] and leads to loose bounds, to bound the mixture of binomial distributions. Moreover, Theorem 3.2 in [23] holds with an assumption on the local privacy budget $\epsilon_0$, which is removed by using $f$-DP in our paper.

     Numerically, we compare our theoretical bound with the numerical upper bound in [23] and the numerical lower bound obtained by privacy auditing in Table 4 in the attached PDF, which shows that our theoretical bound is near-optimal as it closely aligns with the lower bound.

2. The notation $(X|I,I)$ in Section 4 is confusing, and we have addressed this concern in the revised version. Specifically, we clarified the definition as follows: $(X|I, I)$ means we observe two random variables, $X|I$ and $I$, where $X$ follows a mixture distribution $\sum_{i=1}^m w_i p_i$, and $I$ represents the information of indices with $\mathbb{P}[I=i] = w_i$. $X|I$ denotes $X|I=i \sim p_i$. As releasing the indices exposes more information, the couple $((X|I, I), (Y|I, I))$ is more separable than $(X, Y)$ without knowing the indices, which motivates Lemma 4.1.

3. In Section 3.2, we applied Theorem 3.5 to analyze privacy amplification in noiseless 1-dimensional linear models (Example 3.3 and 3.4), where the sensitivity of the gradient $\mu_I$ with a given initialization $I$ can be specified. It is worth noting that similar results are applicable to multidimensional linear models, where $y_i = a^T x_i$ and $x_ix_i^T = \mathbb{I}$ with $\mathbb{I}$ being the identity matrix. Noiseless linear models find relevance in various applications, as exemplified in the work by Berthier et al. (2020) titled "Tight Nonparametric Convergence Rates for Stochastic Gradient Descent under the Noiseless Linear Model."

     For more general applications, such as general convex loss functions, deriving $\mu_I$ becomes more intricate and deserves further investigation, which we plan to address in our future study on the privacy of multi-step DP-SGD.

Overall, the DP-GD part is not as promising as our results on shuffling models but it is still novel with applications to specific examples.

---

### Decision · Program_Chairs · 2023-09-21

**Decision:**

Accept (poster)

**Comment:**

The paper analyzes mixture mechanisms via f-DP and the corresponding tradeoff function. Using this method, the paper improves bounds for shuffled mechanisms for DP and applies it to DP-GD. The paper seems to have good contributions, however reviewers point out several issues including the writing, choice of numbers in experiments. I strongly encourage authors to incorporate all the reviewer comments before the camera ready version.